

# Thalattosuchian crocodylomorphs from European Russia, and new insights into metriorhynchid tooth serration evolution and their palaeolatitudinal distribution

Mark T. Young[1,2,*], Nikolay G. Zverkov[3,*], Maxim S. Arkhangelsky[4,5], Alexey P. Ippolitov[3,6], Igor A. Meleshin[7], Georgy V. Mirantsev[8], Alexey S. Shmakov[8] and Ilya M. Stenshin[9]

[1] School of GeoSciences, University of Edinburgh, Edinburgh, United Kingdom
[2] LWL-Museum für Naturkunde, Münster, Germany
[3] Geological Institute of the Russian Academy of Sciences, Moscow, Russia
[4] Department of General Geology and Minerals, Saratov State University, Saratov, Russia
[5] Department of Oil and Gas, Saratov State Technical University, Saratov, Russia
[6] School of Geography, Environment and Earth Sciences, Victoria University of Wellington | Te Herenga Waka, Wellington, New Zealand
[7] Mordovian Republican United Museum of Local Lore named after I.D. Voronin, Saransk, Republic of Mordovia, Russia
[8] Borissiak Paleontological Institute of the Russian Academy of Sciences, Moscow, Russia
[9] Undory Paleontological Museum, Undory, Ulyanovsk Oblast, Russia
[*] These authors contributed equally to this work.

Corresponding author
Mark T. Young,
marktyoung1984@gmail.com

## ABSTRACT

From the Middle Jurassic to the Early Cretaceous, metriorhynchid crocodylomorphs inhabited marine ecosystems across the European archipelago. Unfortunately, European metriorhynchids are only well known from Germany, France, and the UK, with the Eastern European fossil record being especially poor. This hinders our understanding of metriorhynchid biodiversity across these continuous seaways, and our ability to investigate provincialism. Here we describe eleven isolated tooth crowns and six vertebrae referable to Metriorhynchidae from the Callovian, Oxfordian, Volgian (Tithonian), and Ryazanian (Berriasian) or Valanginian of European Russia. We also describe an indeterminate thalattosuchian tooth from the lower Bajocian of the Volgograd Oblast, the first discovery of a marine reptile from the Bajocian strata of European Russia. These rare fossils, along with previous reports of Russian thalattosuchians, indicate that thalattosuchians have been common in the Middle Russian Sea since it was formed. Palaeolatitude calculations for worldwide metriorhynchid-bearing localities demonstrate that the occurrences in European Russia are the most northern, located mainly between 44–50 degrees north. However, metriorhynchids appear to be rare at these palaeolatitudes, and are absent from palaeolatitudes higher than 50°. These observations support the hypothesis that metriorhynchids evolved an elevated metabolism but were not endo-homeothermic, especially as endo-homeothermic marine reptiles (ichthyosaurs and plesiosaurs) remained abundant at much higher palaeolatitudes.

## INTRODUCTION

During most of the Mesozoic era, the East European Plain was extensively covered by an inland sea, the Middle Russian Sea (Fig. 1B; *Sasonova & Sasonov, 1967*). This inland sea was seemingly a favourable habitat for marine reptiles, as their fossil remains are common in the Jurassic and Cretaceous deposits of European Russia (*e.g.*, *Storrs, Arkhangel'sky & Efimov, 2000*). However, despite the nearly two hundred years of Mesozoic marine reptile research in Russia and numerous finds of ichthyosaurian and plesiosaurian remains, discoveries of thalattosuchian crocodylomorph fossils are still extremely rare (*e.g.*, *Ochev, 1981*; *Hua, Vignaud & Efimov, 1998*; *Storrs & Efimov, 2000*; *Meleshin, 2015*).

Thalattosuchians are a curious group of fossil crocodylomorphs. They evolved from semi-aquatic nearshore predators into fully aquatic forms that lived in open sea environments (*Fraas, 1902*; *Andrews, 1913*; *Buffetaut, 1982*; *Hua & Buffetaut, 1997*; *Young et al., 2010*; *Wilberg, 2015*; *Ősi et al., 2018*; *Schwab et al., 2020*). There are two primary subclades of thalattosuchians: Teleosauroidea, in which the transition to being fully aquatic did not occur (*Buffetaut, 1982*; *Foffa et al., 2019*; *Johnson, Young & Brusatte, 2020*); and Metriorhynchoidea, where the transition to a fully aquatic and pelagic lifestyle did occur (*Fraas, 1902*; *Buffetaut, 1982*; *Hua, 1994*; *Hua & Buffetaut, 1997*; *Young et al., 2010*; *Wilberg, 2015*; *Ősi et al., 2018*). Within the metriorhynchoid subgroup Metriorhynchidae, their aquatic specialisations reached its zenith. Amongst their many pelagic adaptations, metriorhynchids evolved hydrofoil-like forelimbs, a hypocercal tail, loss of bony armour (osteoderms), smooth scaleless skin, hypertrophied salt exocrine glands, and an elevated metabolism (*e.g.*, see *Fraas, 1902*; *Arthaber, 1906*; *Andrews, 1913*; *Buffetaut, 1982*; *Hua, 1994*; *Hua & Buffetaut, 1997*; *Fernández & Gasparini, 2000*; *Fernández & Gasparini, 2008*; *Young et al., 2010*; *Séon et al., 2020*; *Spindler et al., 2021*; *Cowgill et al., 2022*).

Below we describe a set of scattered thalattosuchian remains collected across the territory of European Russian from the Bajocian–Valanginian interval (Fig. 1): an isolated tooth crown from the Bajocian of the Volgograd Oblast; tooth crowns from the Callovian of the Republic of Mordovia, Saratov, Moscow, Ryazan, and Kostroma Oblasts; tooth crowns from the Oxfordian of Vladimir and Moscow Oblasts; isolated cervical, dorsal and caudal vertebrae from the Oxfordian and Volgian of the Ulyanovsk Oblast; and a caudal centrum from the Ryazanian (Berriasian) or Valanginian of Kirov Oblast. Altogether these findings, along with previous reports, indicate that thalattosuchian crocodylomorphs were present in the Middle Russian Sea since the early stages of its development and probably were common in marine reptile assemblages during most of the Middle and Late Jurassic history of this basin.

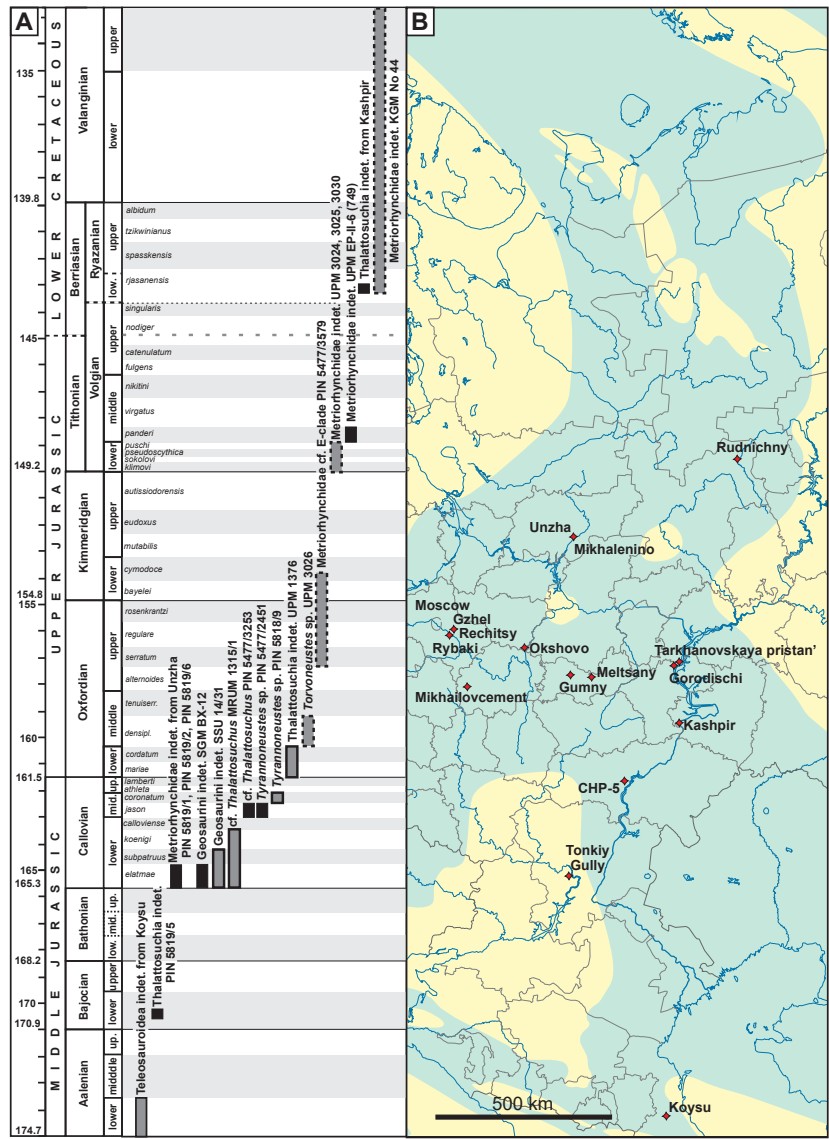

**Figure 1** (A) Stratigraphic distribution of thalattosuchians from European Russia. Specimens with uncertainties in stratigraphic position are shown in grey boxes, specimens collected *ex situ* are shown with dashed margins. (B) Localities of known thalattosuchian fossils from European Russia. Blue coloration on the map outlines of the Middle Russian Sea during the Callovian, based on *Sasonova & Sasonov (1967)*. Chronostratigraphic chart in (A) follows the International Chronostratigraphic Chart (*Cohen et al., 2013*, updated 2023) for international stages and dating of their GSSP, but also considers local stratigraphic units of European Russia and their ammonite zonation when applicable. Ammonite zonation of European Russia on (A) follows *Rogov, Zakharov & Kiselev (2008)* and *Rogov (2021a)*; duration of ammonite zones is approximate and should be taken with caution. The underlying geographic map is based is a modification of the Russian map by Uwe Dedering, available here: https://commons.wikimedia.org/wiki/File:European_Russia_laea_location_map_(without_Crimea).svg.

## Historical background

The first report of presumably thalattosuchian remains from Russia comes from a short communication by *Ochev (1981)*, who reported on a 'posterior dorsal vertebra' and 'fifth metatarsal' that, in his opinion, could belong to a taxon similar to *Dakosaurus* (*Ochev, 1981*). These specimens were collected in the 1930s, *ex situ*, along with remains of Quaternary mammals and some fragments of Jurassic ammonites on the beach of the presently submerged Khoroshovsky Island in the Volga River near Khvalynsk, Saratov Oblast. *Ochev (1981)* suggested that the specimens could be Late Jurassic or Early Cretaceous in age. Subsequent authors have repeatedly cited this report as a thalattosuchian occurrence (*e.g.*, *Efimov, 1988*; *Efimov & Tchkhikvadze, 1987*; *Hua, Vignaud & Efimov, 1998*; *Storrs & Efimov, 2000*; *Young et al., 2010*; *Young et al., 2012a*; *Young et al., 2014a*; *Parrilla-Bel et al., 2013*; *Skutschas, Efimov & Rezvyi, 2015*; *Meleshin, 2015*; *Sachs, Young & Hornung, 2020*; *Madzia et al., 2021*). However, based on our examination of the original specimens deposited in the Khvalynsk museum (MS Arkhangelsky, pers. obs., 2006), we conclude that the vertebra (XKM 1614/1; Fig. S1) is actually a subfossilized Weberian vertebra, and the 'fifth metatarsal' (XKM 1614/2; Fig. S1) is actually the ceratohyale of the catfish *Silurus*. Both specimens likely derived from Quaternary deposits, just like the majority of findings from Khoroshovsky Island (*e.g.*, *Khromov, Arkhangelsky & Ivanov, 2001*). Therefore, the first report of thalattosuchian remains from Russia is erroneous.

The second report of a thalattosuchian from Russia and the only known Aalenian specimen from Russia is an incomplete skull identified as *Steneosaurus* sp. (*Efimov, 1982*; *Efimov, 1988*; *Efimov & Tchkhikvadze, 1987*; *Skutschas, Efimov & Rezvyi, 2015*). The specimen was found in the lower Aalenian of the Karakh Formation at the Koysu locality, Dagestan, Northern Caucasus (*Efimov, 1982*). Given the recent taxonomic and nomenclatural changes of Teleosauroidea introduced by *Johnson, Young & Brusatte (2020)*, this specimen is best considered Teleosauroidea indet.

The subsequent records are associated with the Callovian to Upper Jurassic strata, that are well-represented across the Central part of European Russia. From the Callovian of the Saratov Oblast, a characteristically carinate and lenticular thalattosuchian tooth crown was briefly described and figured by *Arkhangelsky (1999)*; however, it was erroneously attributed to be ichthyosaurian. *Efimov (2010)* depicted an isolated centrum from the Oxfordian of Tetyushi District, Republic of Tatarstan. He identified this specimen (UPM 1376) as belonging to a dinosaur (Ornithischia indeterminate); however, it is most likely a thalattosuchian dorsal centrum (see description of this specimen below). The presence of thalattosuchians in the Callovian and Oxfordian of the Moscow and Ryazan Oblasts was reported by *Zverkov, Shmakov & Arkhangelsky (2017)* in their overview of Jurassic marine reptiles of Moscow and its surroundings: three tooth crowns were identified as Thalattosuchia indet., although their formal description and more accurate identification were deferred for this contribution. Also, *Meleshin (2015)* described an isolated tooth crown, three cm in apicobasal length, from the Oxfordian deposits of Staroye Shaygovo district of the Republic of Mordovia, suggesting its affinity to *Dakosaurus*—unfortunately, this specimen is kept in a private collection and currently unavailable for study.

Besides isolated teeth and skeletal parts, there are two records of articulated (or partly articulated) skeletons. *Hua, Vignaud & Efimov (1998)* described a largely incomplete skeleton of an indeterminate metriorhynchid (UPM EP-II-6 (749)) from the middle Volgian (Volgian Regional Stage, which largely corresponds to the Tithonian International Stage) of Gorodischi locality in Ulyanovsk Oblast. In 2015, the discovery of a bone association assumed to be that of a metriorhynchid was announced and figured by Efimov and Meleshin (*Efimov, 2015*; *Meleshin, 2015*). This specimen, initially found by MA Rogov, is from the Ryazanian regional stage (*Riasanites rjasanensis* ammonite Biozone, earliest Cretaceous) of the Kashpir locality, Samara Oblast. Currently, this unique find is in the private collection of V.M. Efimov and is still under his study.

## GEOLOGICAL SETTINGS AND PALEOGEOGRAPHY

The marine transgression onto the East European Plain started in the late Early–early Middle Jurassic, at first as progressively widening bays in the south and north. The oldest thalattosuchian find described in the present paper, PIN 5819/5, was collected from the lower Bajocian strata of the Volgograd Oblast, represented by silt-clay intercalation, and palaeogeographially relates to the southern bay. By the beginning of the Bathonian, the marine transgression had reached the central regions of the East European Plain, forming a short-lived epicontinental marine seaway that connected the Tethys Ocean with the Arctic seas (*e.g.*, *Sasonova & Sasonov, 1967*; *Mitta et al., 2004*; *Ippolitov, 2018a*; *Ippolitov & Desai, 2019*). Unfortunately, there are no records of thalattosuchians from the lower Bathonian of Central Russia, which is relatively well-studied, probably due to the prominent cold-water inflow this time from the Arctic, as can be supposed from the observed invertebrate immigration from high latitudes (*Mitta et al., 2004*; *Ippolitov, 2018a*).

The Middle Bathonian was a time of significant regression. The new transgression cycle started from the late Bathonian, and by the early Callovian, the Middle Russian Sea covered the East European Plain very extensively. At this time, the shallow marine basin permanently covered large territories, and traces of this sea are reflected in multiple available fossil localities. There are six lower Callovian specimens (PIN 5819/1, PIN 5819/2, PIN 5819/6, SGM BX-12, MRUM 1315/1, SSU 14/31) described herein, originating from three different localities in three different regions (Kostroma Oblast, Saratov Oblast, Republic of Mordovia), all attributed to its lower half; and the middle Callovian record includes four specimens (PIN 5819/7, PIN 5818/9, PIN 5477/3253, PIN 5477/2451) from two localities in neighboring regions (Moscow and Ryazan Oblasts; Fig. 1B).

The overlying Late Jurassic strata of Oxfordian, Kimmeridgian, and early to middle Volgian (Tithonian) ages reflect the continuations of a significant transgressional phase reaching its maximum in the middle Volgian (*Alekseev & Olferiev, 2007*). Strata of these ages are normally characterized by clay facies, representing more offshore environments, while nearshore deposits were later eroded and currently are poorly represented in European Russia. This interval also contains some thalattosuchian records, however, much fewer than in the Callovian: there is a find from the upper Oxfordian–lower Kimmeridgian clays

of Moscow Oblast and another one from the poorly known locality of middle Oxfordian age in Vladimir Oblast.

The succession near Gorodischi village (25 km north of Ulyanovsk) represents one of the best sections across the upper part of the Upper Jurassic in Central Russia and was selected as the lectostratotype of the Volgian stage (*Gerasimov & Mikhailov, 1966*). The outcrop here ranges from the Kimmeridgian *Aulacostephanus eudoxus* to the upper Volgian *Craspedites nodiger* ammonite Biozones (*e.g.*, *Price & Rogov, 2009*; *Rogov, 2010*; *Rogov, 2021a*). A single specimen UPM EP-II-6 (749) was found in the black shales of the middle Volgian *Dorsoplanites panderi* ammonite Biozone (*Hua, Vignaud & Efimov, 1998*), while four more specimens have all been collected *ex situ* on the riverbank; however, their preservation suggests that they most likely originate from either the lower Volgian strata (specimens UPM 3024, 3025, 3030) or middle Volgian *Dorsoplanites panderi* ammonite Biozone (strongly pyritized dorsal vertebra UPM 3031).

The upper Volgian to Valanginian succession forms a major regressive series and is usually characterized by strongly condensed, extremely shallow-water facies. The stratigraphically youngest specimen described herein, KGM No 44, comes from the lowermost Cretaceous strata of Rudnichny quarry, Kirov Oblast. Considering the lithostratigraphy of this quarry (see *Morozov et al., 1967*; *Zverkov et al., 2018*), the age of the specimen can conservatively be determined as being Ryazanian (Berriasian) or Valanginian, although the preservation of KGM No 44 is most similar to pliosaurid teeth from the Ryazanian beds of this locality (*Zverkov et al., 2018*).

## MATERIALS & METHODS

The material examined as part of the present study is summarized in Table 1. Teeth were coated in ammonium chloride prior to being photographed; photographs without coating are provided as supplementary materials. Additionally, specimens were studied using Scanning electron microscopes (SEM) TESCAN Vega 2 and TESCAN Vega 3 in the Borissiak Paleontological Institute of RAS.

We used present-day GPS coordinates for thalattosuchian-bearing localities (Table S1) to calculate palaeolatitudes by applying the Paleolatitude.org online calculator (*van Hinsbergen et al., 2015*) and its default palaeomagnetic frame (*Torsvik et al., 2012*). The resulting palaeolatitude calculations are presented in Table 2 and Table S1.

The nomenclature of biostratigraphic units follows the International Stratigraphic Guide (*Murphy & Salvador, 1999*). The Formation names, which are often omitted in the biostratigraphical literature on the region, non-critically follow Unified regional stratigraphic schemes (*Chirva, 1993*; *Mitta, 2012*), except otherwise stated.

Young et al. (2023), *PeerJ*, DOI 10.7717/peerj.15781

Peerj

**Table 1   List of known Russian thalattosuchian fossils.**

| Specimen number and institution | Material | Historical taxonomic identification | Taxonomic identification herein | Age | Locality | Reference |
|---|---|---|---|---|---|---|
| unknown | skull | *Steneosaurus* sp. | Teleosauroidea indet. | early Aalenian, Middle Jurassic | Koysu locality, Dagestan, Northern Caucasus | *Efimov (1982)*; *Efimov (1988)*; *Skutschas, Efimov & Rezvyi (2015)* |
| PIN 5819/5 | tooth crown | – | Thalattosuchia indet. | early Bajocian, Middle Jurassic | Tonkiy Gully (''Tonkiy Yar''), Ilovlya district, Volgograd Oblast | Herein |
| PIN 5819/1 | tooth crown | – | Metriorhynchidae indet. | early Callovian, Middle Jurassic | Unzha River, Makariev District, Kostroma Region | Herein |
| PIN 5819/2 | tooth | – | Metriorhynchidae indet. | early Callovian, Middle Jurassic | Unzha River, Makariev District, Kostroma Region | Herein |
| SGM BX-12 | tooth crown | – | Geosaurini indet. Morphotype 2 | early Callovian, Middle Jurassic | Unzha River, Makariev District, Kostroma Region | Herein |
| SSU 14/31 (104a/29) | tooth crown | 'tooth of a latipinnate ichthyosaur' | Geosaurini indet. Morphotype 1 | early Callovian, Middle Jurassic | CHP 5 power station, Saratov, Saratov Oblast, Russia | *Arkhangelsky (1999)* |
| MRUM 1315/1 | Tooth crown | – | cf. *Thalattosuchus* | early Callovian, Middle Jurassic | Gumny village, Krasnoslobodsk district of the Republic of Mordovia. | Herein |
| PIN 5818/9 | tooth crown | – | *Tyrannoneustes* sp. | middle Callovian, Middle Jurassic | Rechitsy Village, Ramenskoe District, Moscow Region | Herein |

Young et al. (2023), *PeerJ*, DOI 10.7717/peerj.15781

**Table 1** (*continued*)

| Specimen number and institution | Material | Historical taxonomic identification | Taxonomic identification herein | Age | Locality | Reference |
|---|---|---|---|---|---|---|
| PIN 5477/2451 | tooth crown | Thalattosuchia indet. | *Tyrannoneustes* sp. | middle Callovian, Middle Jurassic | Mikhailovcement Quarry, Ryazan region | *Zverkov, Shmakov & Arkhangelsky (2017)* |
| PIN 5477/3253 | tooth crown | Thalattosuchia indet. | cf. *Thalattosuchus* | middle Callovian, Middle Jurassic | Mikhailovcement Quarry, Ryazan region | *Zverkov, Shmakov & Arkhangelsky (2017)* |
| UPM 1376[*] | Dorsal centrum | Ornitischia indet. (*Efimov, 2010*) | Thalattosuchia indet. | early Oxfordian, Late Jurassic | Tarkhanovskaya pristan, Tetyushi District, Tatarstan Region | *Efimov (2010)* |
| UPM 3026 | Tooth crown | – | *Torvoneustes* sp. | middle Oxfordian, Late Jurassic | Okshovo village, Melenki District, Vladimir region | Herein |
| Private collection | Tooth crown | cf. *Dakosaurus* | Geosaurini indet. cf. E-clade | Oxfordian, Late Jurassic | Meltsany village, Staroe Shaigovo district of the Republic of Mordovia | *Meleshin (2015)* |
| PIN 5477/3579 | tooth crown | Thalattosuchia indet. | Geosaurini indet. cf. E-clade | late Oxfordian –?earliest Kimmeridgian, Late Jurassic | Rybaki village, Ramenskoe District, Moscow region | *Zverkov, Shmakov & Arkhangelsky (2017)* |
| XKM 1614/1 & 1614/2 | Dorsal vertebra | cf. *Dakosaurus* (*Ochev, 1981*) | Weberian vertebra of *Silurus* (extant catfish) | Ex situ, was suggested to be Late Jurassic in original description, but is associated with Quaternary mammal fauna | Khoroshovsky Island on the Volga River near Khvalynsk, Saratov Region | *Ochev (1981)* |
| UPM EP-II-6 (749)[*] | Fragmentary skeleton | Metriorhynchidae indet. | Metriorhynchidae indet. | middle Volgian (=early Tithonian), Late Jurassic | Gorodischi locality in Ulyanovsk Region | *Hua, Vignaud & Efimov (1998)* |

Young et al. (2023), *PeerJ*, DOI 10.7717/peerj.15781

**Table 1** (*continued*)

| Specimen number and institution | Material | Historical taxonomic identification | Taxonomic identification herein | Age | Locality | Reference |
|---|---|---|---|---|---|---|
| UPM 3024 and UPM 3025 | Cervical and caudal vertebrae | – | Metriorhynchidae indet. | early-to-middle Volgian (=early Tithonian), Late Jurassic | Gorodischi locality in Ulyanovsk Region | Herein |
| Uncatalogued. Currently in the private collection of VM Efimov | Fragmentary skeleton | 'marine crocodile' | Thalattosuchia indet. | Ryazanian (=late Berriasian), Early Cretaceous | Kashpir locality, Samara Region | *Efimov (2015)*; *Meleshin (2015)* |
| KGM No 44 | Dorsal vertebra centrum | – | Metriorhynchidae indet. | Berriasian or Valanginian, Early Cretaceous | Vyatka-Kama phosphate field, Verkhnekamsky District, Kirov Region | Herein |

**Notes.**

*Note, these specimens were not found to be present in the UPM collection after the retirement of VM Efimov in 2018.

# SYSTEMATIC PALEONTOLOGY

CROCODYLOMORPHA *Hay, 1930* (sensu *Nesbitt, 2011*)
THALATTOSUCHIA *Fraas, 1901* (sensu *Young & Andrade, 2009*)
THALATTOSUCHIA INDET. (Fig. 2; Fig. S2)

**Specimen**—PIN 5819/5, an incomplete isolated tooth crown. Specimen was collected by A.P. Ippolitov.

**Locality**—Tonkiy Gully near the Dubovoi Hamlet, Ilovlya District, Volgograd Oblast, Russia (for details see *Ippolitov, 2018b*).

**Horizon**—680 cm above the base of Member II; beds with *Hastites orphana* (belemnite-based unit, equivalent of the *Witchellia laeviuscula* ammonite Chronozone), lower Bajocian, Middle Jurassic, Bakhtemir Formation (*sensu Ippolitov, 2018b*).

**Description**—The tooth crown is incomplete, with the apical region missing and appears to be worn, and the basal region seemingly broken near the root-crown junction (Fig. 2). The basal region is poorly preserved, with much of the enamel and underlying dentine broken or eroded (Figs. 2A–2B, 2D–2E). The incomplete apex shows the underlying dentine, which has a smoothed surface (Figs. 2C, 2H). The crown is poorly preserved, particularly in the basal region, and a small portion of the root is present. There is a pronounced break in approximately one-quarter of the way from the preserved basal region, with the central labial region of the crown badly damaged at this break (Figs. 2A, 2D–2E). From what is preserved, the tooth crown would have had a subconical shape, being both lingually curved and mediolaterally compressed. The labial surface lacks both apicobasal facets (see *Young & Andrade, 2009*; *Andrade et al., 2010*; *Foffa et al., 2018a*; *Herrera, Aiglstorfer & Bronzati, 2021*) and apicobasal fluting (see *Foffa et al., 2018a*). Based on its relatively short apicobasal height and the breadth of the basal region, it was perhaps from the posterior end of the tooth row. We cannot determine whether it came from the upper or lower tooth row.

Mesial and distal carinae are present in PIN 5819/5 and are formed by a carinal keel (raised ridge). The keels are very prominent, as in the early-diverging metriorhynchoids *Magyarosuchus fitosi* and *Zoneait nargorum* (*Wilberg, 2015*; *Ősi et al., 2018*). Although prominent, they lack 'carinal flanges' (*i.e.*, when the crown becomes concave immediately adjacent to the carinae; *Chiarenza et al., 2015*; *Young et al., 2015a*). There are no serrations present along the keel, either 'false serrations' created by the superficial enamel ornamentation contacting the carinal keel or discrete denticles (Figs. 2D–2I). The carinae are somewhat 'wavy' in places giving the impression of denticles (see Fig. 2F). As such, we can exclude PIN 5819/5 from the teleosauroid subclade Machimosaurini and the metriorhynchid subclade Geosaurinae (*Andrade et al., 2010*; *Young et al., 2013a*; *Young et al., 2014b*; *Young et al., 2015b*; *Foffa et al., 2018a*; *Foffa et al., 2018b*).

The preserved external enamel surfaces are covered with numerous apicobasally aligned ridges that are arranged in a (sub)-parallel manner. Most of the enamel ridges are continuous from the basal region to the preserved apical-most region, although shorter ridges are also present, as are ridges that contact and fuse. The enamel ridges are more

**Table 2** Palaeolatitudinal and paleotemperature estimates for the metriorhynchid specimens described herein. Palaeolatitudes were calculated using paleolatitude.org.

| Specimen and identification | Geography | Age | Coordinates | Paleotemperature estimates and reference |
|---|---|---|---|---|
| Metriorhynchidae indet. (PIN 5819/1, 5819/2 and 5819/6) and Geosaurini indet. morphotype 2 (SGM BX-12) | Unzha village, Makariev district, Kostroma Oblast | *Cadoceras elatmae* chron, early Callovian, Middle Jurassic | 57.9905, 44.0019 (50N for 170–160 Ma) | ~5–13 °C (*Wierzbowski et al., 2020*) |
| Geosaurini indet. morphotype 1 (SSU 14/31) | CHP-5, Saratov, Saratov Oblast | *Cadoceras elatmae–Cadochamoussetia subpatruus* chrons, early Callovian, Middle Jurassic | 51.6240, 45.9899 (45N for 170–160 Ma) | 5.5–11 °C Boreal fauna present (*Wierzbowski et al., 2020*) |
| cf. *Thalattosuchus* (MRUM 1315/1) | Gumny village, Krasnoslobodsk district, the Republic of Mordovia. | *Cadoceras elatmae–Proplanulites koenigi* chrons, early Callovian, Middle Jurassic | 54.389, 43.728 (47N for 170–160 Ma) | 8.5–13 °C (*Wierzbowski et al., 2020*) |
| *Tyrannoneustes* sp. (PIN 5818/9) and Metriorhynchidae indet. (PIN 5819/7) | Rechitsy village, Ramenskoe district, Moscow Oblast | *Erymnoceras coronatum* chron, middle Callovian, Middle Jurassic | 55.60, 38.44 (47N for 170–160 Ma) | No palaeotemperature estimates available. Considered thermal maximum with multiple Tethys molluscs (*Rogov, Zakharov & Kiselev, 2008*) |
| cf. *Thalattosuchus* (PIN 5477/3253) and *Tyrannoneustes* sp. (PIN 5477/2451) | Mikhaylovcement quarry, Mikhaylov district, Ryazan Oblast | *Kosmoceras jason* chron, middle Callovian, Middle Jurassic | 54.2112, 38.9373 (46N for 170–160 Ma) | 6.3–10.5 °C Boreal fauna present (*Wierzbowski et al., 2020*) |
| Thalattosuchia indet. (UPM 1376) | Tarkhanovskaya pristan, Tetyushi district, Republic of Tatarstan | early Oxfordian, Late Jurassic | 54.655, 48.603 (49N for 160 Ma) | 13.9–19.5 °C (*Wierzbowski et al., 2018*) |
| *Torvoneustes* sp. (UPM 3026) | Okshovo village, Melenki district, Vladimir Oblast | *Cardioceras densiplicatum* chron, middle Oxfordian, Late Jurassic | 55.134, 41.723 (47N for 160 Ma) | 10–21 °C (*Wierzbowski et al., 2018*) |
| Metriorhynchidae cf. 'E'-clade (PIN 5477/3579) | Rybaki village, Ramenskoe district, Moscow Oblast | late Oxfordian or earliest Kimmeridgian, Late Jurassic | 55.4804, 38.2173 (46N for 160-150 Ma) | 13.7–22.3 °C (*Wierzbowski et al., 2018*) |
| Metriorhynchidae indet. (UPM 3031) | Gorodischi locality, Ulyanovsk district, Ulyanovsk Oblast | *Dorsoplanites panderi* chron, middle Volgian, Late Jurassic | 54.5765, 48.4176 (47N for 150 Ma) | 18–20 °C (*Ruffell et al., 2002*; (*Gröcke et al., 2003*; *Price & Rogov, 2009*) |
| Thalattosuchia indet. (*Efimov, 2015*) | Kashpir locality, Samara Oblast | *Riasanites rjasanensis* chron, Berriasian, Early Cretaceous | 53.0282, 48.4532 (38N for 140 Ma) | 15.5–18.3 °C (*Gröcke et al., 2003*) |
| Metriorhynchidae indet. (KGM No 44) | Vyatka-Kama phosphate field, Verkhnekamsky District, Kirov Oblast | Berriasian or Valanginian, Early Cretaceous | 59.59, 52.49 (45N for 140 Ma) | ~15.5–19 °C (Based on Kashpir, (*Gröcke et al., 2003*)) |

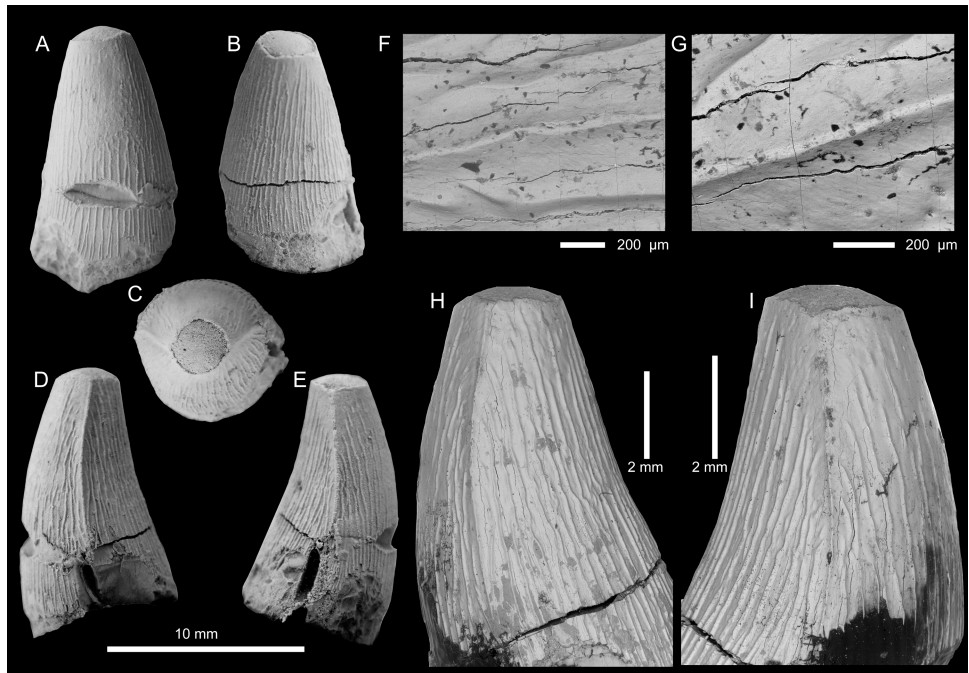

**Figure 2** **Thalattosuchia indet. tooth crown (PIN 5819/5) from the lower Bajocian of Tonkiy Gully, Volgograd Oblast, Russia.** (A–E) Tooth crown in labial (A), lingual (B), apical (C), and mesial/distal (D, E) views. F, G, SEM photographs of the carina. Magnified SEM photographs of (C) and (D) are (H) and (I) respectively.

regular on the lingual surface than the labial one. On the labial surface, the ornamentation is less regular, particularly in the central region closer to the preserved apical part. There, there is a region where the ornamentation becomes faint, and there are shorter ridges and ridges which become less defined. A shift to a smoother enamel ornamentation on the labial surface also occurs in the early-diverging metriorhynchoids *Pelagosaurus typus* and *Opisuchus meieri* (*Aiglstorfer, Havlik & Herrera, 2020*). PIN 5819/5 differs from the early-diverging metriorhynchoid *Teleidosaurus calvadosii*, as that taxon has less ornamented dentition and crowns which have apicobasal fluting (see *Hua, 2020*). PIN 5819/5 also differs from the early-diverging metriorhynchoid *Magyarosuchus fitosi*, as in that taxon the apicobasal ridges continue to the apex (*Ősi et al., 2018*). In both *Magyarosuchus fitosi* and *Zoneait nargorum* the carinae do not reach the basal region of the crowns (*Wilberg, 2015*; *Ősi et al., 2018*), however, we cannot ascertain whether the same was true for PIN 5819/5, due to the damage to the tooth crown.

THALATTOSUCHIA *Fraas, 1901* (sensu *Young & Andrade, 2009*)
METRIORHYNCHIDAE *Fitzinger, 1843* (sensu *Young & Andrade, 2009*)
METRIORHYNCHIDAE INDET. Morphotype 1 (Figs. 3A–3S; Fig. S4)

**Specimens**—PIN 5819/1, PIN 5819/2, and PIN 5819/6, isolated tooth crowns. Specimens collected by A.V. Stupachenko.

**Locality**—Unzha Village, Makariev District, Kostroma Oblast, Russia (for details see *Mitta, 2000*).

**Horizon**— *Cadoceras elatmae* ammonite Biozone; lower Callovian, Middle Jurassic, Kologriv Formation.

**Description**—The three tooth crowns are all incomplete, with the apices missing and extensive damage to the enamel on the labial and lingual surfaces (Figs. 3A–3S). At least part of the root is preserved for all three crowns, with PIN 5819/6 having the most complete root (Figs. 3N–3R). Based on what is preserved of all three crowns, they would have been subconical in shape, being mediolaterally compressed, curved lingually and slightly distally. There is no evidence on their labial surfaces of apicobasal facets (see *Young & Andrade, 2009*; *Andrade et al., 2010*; *Foffa et al., 2018a*; *Herrera, Fernández & Vennari, 2021*) or apicobasal fluting (see *Foffa et al., 2018a*).

Mesial and distal carinae are present in all three crowns, albeit highly incomplete. The carinae for all three crowns are highly damaged and appear to have suffered taphonomic wear, making them 'rounded' (Figs. 3F, 3L, 3M, 3S). A keel is present but we cannot make any definitive statements on the morphology of the carinal keel or whether there were any denticles present. Under scanning electron microscopy, no denticles can be reliably observed (Figs. 3F, 3L, 3M, 3S). Based on what is preserved of the carinae in the three crowns, the superficial enamel ornamentation does not contact the keel (= no false serrations), and the keels are continuous along the crown. The is also no evidence for 'carinal flanges' (see *Chiarenza et al., 2015*; *Young et al., 2015a*)

Both the labial and lingual surfaces of all three crowns are well ornamented (Figs. 3A–3E, 3G–3K, 3N–3R). Both surfaces have numerous, elongate apicobasally aligned ridges. The ridges are discontinuous and show great variability in length. Overall, the ridges are more elongated in PIN 5819/6 (Figs. 3N–3R) than PIN 5819/1 (Figs. 3A–3E) and PIN 5819/2 (Figs. 3G–3K). The enamel ridges on PIN 5819/1 (Figs. 3A–3E) are the proportionally shortest and also the most closely packed, especially on the lingual surface. In *Thalattosuchus superciliosus*, newly erupted tooth crowns have a more intense ornamentation pattern composed of ridges being more closely packed (*Young et al., 2013a*). Therefore, the variation amongst these teeth in ridge length and spacing could be the result on natural variation within one species and not have systematic importance.

Middle Jurassic teleosauroids from Western Europe have two distinct dental morphologies (*Vignaud, 1997*), none of which these tooth crowns match. The first, Type A of *Vignaud (1997)* is found in longirostrine forms, and is characterised by a strong lingual curvature, pointed apex, crown being labiolingually 'thin' or 'slender', the crown is more than 2.7 times longer apicobasally than labiolingually wide, the enamel ornamentation is composed of continuous ridges that never reaches the apical region even in replacement teeth. Type B of *Vignaud (1997)* are found in mesorostrine forms (Machimosaurinae sensu *Johnson, Young & Brusatte, 2020*), and are characterised by being more 'robust', with a blunter apex, more intense enamel ornamentation composed of numerous ridges of high relief that anastomose in the apical region, the crown apicobasal length to labiolingual width

ratio is 2.5 or less. The tooth crowns described above do not match either morphotype, being neither strongly curved lingually nor robust, and the enamel ornamentation is composed of low relief ridges that are highly variable in length.

Given the lack of apicobasal faceting and fluting on the labial surface, and 'carinal flanges', these tooth crowns cannot be from members of Geosaurina, Plesiosuchina, or the *Dakosaurus*-lineage (*i.e.,* derived geosaurine lineages; *Andrade et al., 2010*; *Young et al., 2013a*; *Foffa et al., 2018a*; *Foffa et al., 2018b*). Crown shape and enamel ornamentation for all three crowns are consistent with *Thalattosuchus* (*Young et al., 2013a*). However, we cannot assess the morphology of the carinae reliably, and none of the crowns preserve any other diagnostic features. Therefore, we identify all three tooth crowns as Metriorhynchidae indet.

METRIORHYNCHIDAE INDET. Morphotype 2 (Figs. 3T–3Y; Fig. S3)

**Specimen—**PIN 5819/7, poorly preserved tooth crown. Collector is unknown, probably collected around 1970.

**Locality—** Gzhel, near Rechitsy Village, Ramenskoe District, Moscow Oblast, Russia (for details on local geology see *Gerasimov et al., 1996*).

**Horizon—**middle to lower upper Callovian, Middle Jurassic, Kriusha Formation.

**Description—**The tooth crown is incomplete. The majority of the enamel is not preserved, and the apex is missing. It is unclear whether the crown-root junction is preserved, but there is a constriction at the base that could be the junction. From what is preserved, the tooth crown was poorly curved lingually but noticeably curved distally. The crown was also mediolaterally compressed. Based on what is preserved of the enamel, there were no apicobasal facets (see *Young & Andrade, 2009*; *Andrade et al., 2010*; *Foffa et al., 2018a*; *Herrera, Fernández & Vennari, 2021*) or apicobasal fluting (see *Foffa et al., 2018a*) on the labial surface. Given what little of the enamel is preserved, it is hard to discern what patterns there were, although there are some apicobasal ridges on the lingual surface (Fig. 3U). The carinae cannot be described.

Based on the preserved material (lacking well defined enamel ridges and a noticeable lingual curvature), we cannot refer the crown to either teleosauroid dental morphotype (*Vignaud, 1997*). As the crown lacks both apicobasal facets and fluting, and the ornamentation is macroscopically 'smooth', we can exclude *Ieldraan melkshamensis* (*Foffa et al., 2018a*) as a potential identification. The largely smooth enamel ornamentation allows us to exclude *Thalattosuchus superciliosus* (*Young et al., 2013a*), and some Late Jurassic metriorhynchid lineages, such as *Torvoneustes* (*Andrade et al., 2010*; *Young et al., 2013a*; *Young et al., 2013b*; *Young et al., 2020a*; *Barrientos-Lara et al., 2016*) and the 'E'-clade (*Abel, Sachs & Young, 2020*) as potential sources for this crown. Given the preservation cannot determine whether this crown is from *Tyrannoneustes*.

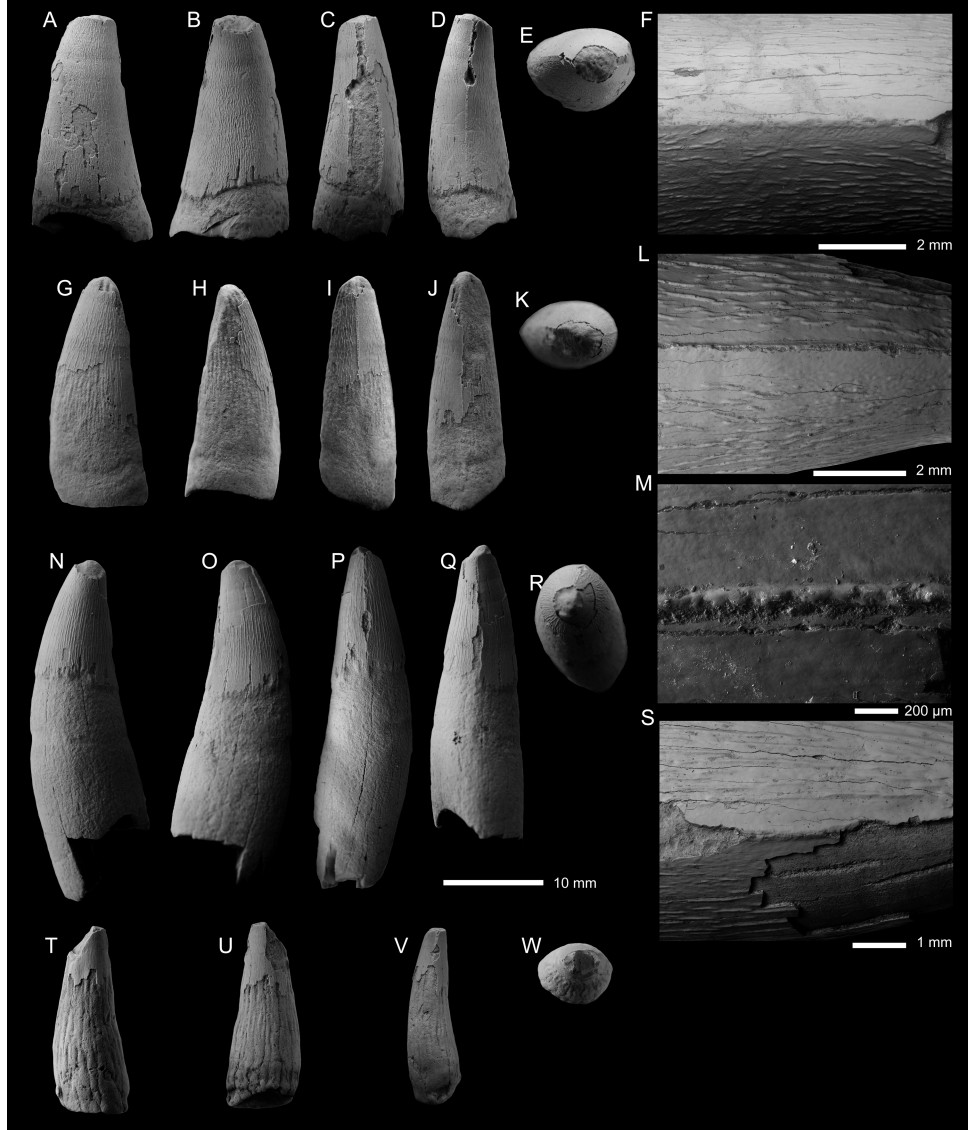

**Figure 3** **Metriorhynchid tooth crowns from the lower and middle Callovian.** (A–F) Metriorhynchidae indet. (PIN 5819/6) from the lower Callovian of the Unzha River, Kostroma Oblast, Russa. (G–M) Metriorhynchidae indet. (PIN 5819/1) from the lower Callovian of the Unzha River, Kostroma Oblast, Russia. (N–S) Metriorhynchidae indet. (PIN 5819/2) from the lowerCallovian of the Unzha River, Kostroma Oblast, Russia. (T–W) Metriorhynchidae indet. (PIN 5819/7) from the middle Callovian of Gzhel Village, Ramenskoe District, Moscow Oblast, Russia. Teeth are depicted in labial (A, G, N, T), lingual (B, H, O, U), mesial (C, I, P), distal (D, J, Q, V), and apical (E, K, R, W) views. F, L, M, S are SEM photographs of carinae.

METRIORHYNCHINAE *Fitzinger, 1843* (sensu *Young & Andrade, 2009*)
cf. THALATTOSUCHUS (Fig. 4; Fig. S5)

**Specimen**—MRUM 1315/1, tooth crown. Specimen collected by I.A. Meleshin.
**Locality**—Gumny village, Krasnoslobodsk district, Republic of Mordovia, Russia.

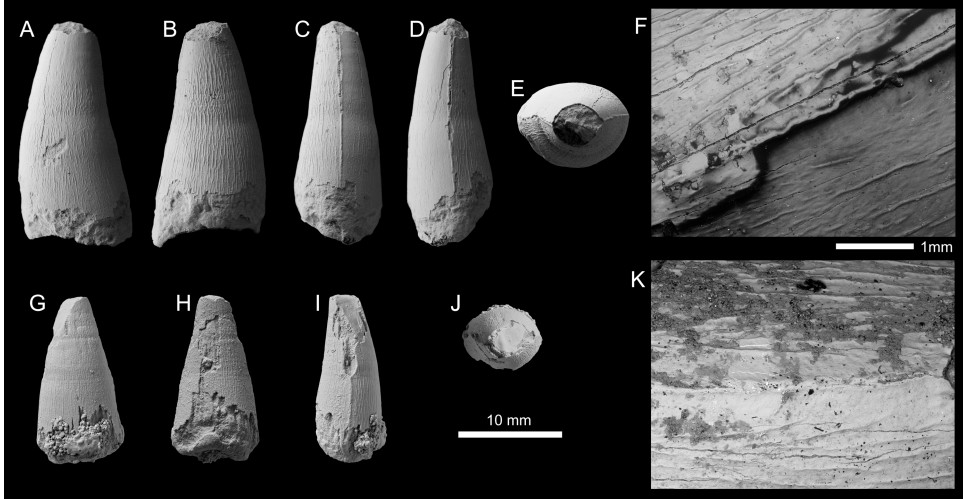

**Figure 4** **Metriorhynchid tooth crowns of cf. *Thalattosuchus* from the lower and middle Callovian of European Russia.** (A–F) cf. *Thalattosuchus* (MRUM 1315/1) from Gumny village, Republic of Mordovia, Russia; (G–K) cf. *Thalattosuchus* (PIN 5477/3253) from the middle Callovian, Mikhaylovcement Quarry, Ryazan Oblast, Russia. Crowns are depicted in labial (A, G), lingual (B, H), mesial (C, I), distal (D), and apical (E, J) views. F, K are SEM photographs of carinae.

**Horizon**—lower Callovian, Middle Jurassic, upper part of the ?Yelatma Formation. There is no clear evidence of the precise age; however, the specimen was collected from sandy facies, which cover the interval from *Cadoceras elatmae* to *Proplanulites koenigi* ammonite Biozones in the local succession (AP Ippolitov, pers. obs., Summer 2019).

**Description**—The tooth crown of MRUM 1315/1 is incomplete, the apex is missing, and there is damage to the enamel on the labial surface and on both carinae (Figs. 4A–4F). Due to damage, it is unclear whether the root-crown junction is preserved (Figs. 4A–4D). The crown has a subconical shape, being mediolaterally compressed, curved lingually and poorly curved distally. The labial surface lacks apicobasal facets (see *Young & Andrade, 2009*; *Andrade et al., 2010*; *Foffa et al., 2018a*; *Herrera, Fernández & Vennari, 2021*) and apicobasal fluting (see *Foffa et al., 2018a*).

Mesial and distal carinae are present in MRUM 1315/1 and are formed by a carinal keel. The keels are continuous on what is preserved on the crown. The carinae have a somewhat 'thick' appearance, caused by the keel being poorly tapered (Figs. 4C–4F), much like that seen in *Thalattosuchus superciliosus* (see figure 23 in *Young et al., 2013a*). This is in contrast to the carinal keel of taxa like *Gracilineustes leedsi*, where the keels are strongly tapered (see figure 24 in *Young et al., 2013a*). The carinae lack 'carinal flanges' (see *Chiarenza et al., 2015*; *Young et al., 2015a*) and have no serrations. Under scanning electron microscopy, there are no denticles, even poorly defined microscopic ones (Fig. 4F), and the superficial enamel ornamentation does not contact the keel (= no false serrations). The carinae do become conspicuously 'wavy' in places (Fig. 4F).

Both the labial and lingual surfaces are well ornamented (Figs. 4A–4E). Apically on the labial surface, the crown becomes more macroscopically 'smooth' (Figs. 4A, 4E). Both

surfaces have numerous, elongate apicobasally aligned ridges. The ridges are discontinuous and show great variability in length. However, the ridges are consistently more elongated and more closely packed on the lingual surface than on the labial surface (*i.e.,* more heavily ornamented, Figs. 4A, 4B).

Given the lack of serrations, apicobasal faceting and fluting on the labial surface, and 'carinal flanges', MRUM 1315/1 can be excluded from Geosaurinae (*Andrade et al., 2010*; *Young et al., 2013a*; *Foffa et al., 2018a*; *Foffa et al., 2018b*). The 'thick' carinae, and the presence of numerous, elongate apicobasal enamel ridges on the labial and lingual surfaces are consistent with *Thalattosuchus* (*Young et al., 2013a*). As is the shift towards a macroscopically 'smoother' apical region (*Young et al., 2013a*). Therefore, we identify MRUM 1315/1 as cf. *Thalattosuchus*.

**Specimen**—PIN 5477/3253, tooth crown. Specimen collected by A.S. Shmakov.

**Locality**—Mikhaylovcement Quarry, Mikhaylov District, Ryazan Oblast, Russia (for details see *Kiselev & Rogov, 2018*).

**Horizon**—*Kosmoceras jason* ammonite Biozone, middle Callovian, Middle Jurassic, Chulkovo Formation.

**Description**—The tooth crown is largely incomplete: the apex is missing, along with much of the apical labial surface (Figs. 4G–4J). Due to damage, it is unclear whether the root-crown junction is preserved (Figs. 4G–4I), basally the crown is badly broken, and much of the enamel is missing (particularly on the lingual surface; Fig. 4H). Based on what is preserved of the crown, it had a subconical shape, being mediolaterally compressed and curved lingually. There is no evidence that the labial surface had apicobasal facets (see *Young & Andrade, 2009*; *Andrade et al., 2010*; *Foffa et al., 2018a*; *Herrera, Fernández & Vennari, 2021*) or apicobasal fluting (see *Foffa et al., 2018a*).

Mesial and distal carinae are present in PIN 5477/3253 and are formed by a carinal keel. The keels are continuous on what is preserved on the crown. The carinae have a somewhat 'thick' appearance, caused by the keel being poorly tapered (Figs. 4I, 4K), much like that seen in *Thalattosuchus superciliosus* (see figure 23 in *Young et al., 2013a*). This is in contrast to the carinal keel of taxa like *Gracilineustes leedsi*, where the keels are strongly tapered (see figure 24 in *Young et al., 2013a*). The carinae lack 'carinal flanges' (see *Chiarenza et al., 2015*; *Young et al., 2015a*) and have no serrations. Under scanning electron microscopy, there are no denticles, even poorly defined microscopic ones (Fig. 4K), and the superficial enamel ornamentation does not contact the keel (= no false serrations).

Both the labial and lingual surfaces are well ornamented (Figs. 4G, 4H). Apically on the labial surface, the crown becomes more macroscopically 'smooth' (Figs. 4G, 4J); unfortunately too much of the crown is missing to assess any other differences. Both surfaces have numerous, elongate apicobasally aligned ridges. The ridges are discontinuous and show great variability in length. On the labial surface, some of the ridges bend towards the carina but do not contact it (Figs. 4I, 4K).

Given the lack of serrations, apicobasal faceting and fluting on the labial surface, and 'carinal flanges', PIN 5477/3253 can be excluded from Geosaurinae (*Andrade et al., 2010*; *Young et al., 2013a*; *Foffa et al., 2018a*; *Foffa et al., 2018b*). The 'thick' carinae, and the presence of numerous, elongate apicobasal enamel ridges on the labial and lingual

surfaces are consistent with *Thalattosuchus* (*Young et al., 2013a*). As is the shift towards a macroscopically 'smoother' apical region (*Young et al., 2013a*). Therefore, we identify PIN 5477/3253 as cf. *Thalattosuchus*.

GEOSAURINAE *Bonaparte, 1845* (sensu *Young & Andrade, 2009*)
TYRANNONEUSTES sp. (Fig. 5; Fig. S6, S7)

**Specimen**—PIN 5477/2451, tooth crown. Specimen collected by A.S. Shmakov.
**Locality**—Mikhaylovcement Quarry, Mikhaylov District, Ryazan Oblast, Russia (for details see *Kiselev & Rogov, 2018*).
**Horizon**—*Kosmoceras jason* Ammonite Biozone, middle Callovian, Middle Jurassic, Chulkovo Formation.
**Description**—The tooth crown is largely complete, with only the tip of the apex broken (Figs. 5A–5E). There is some damage to the carinae (Figs. 5C, 5E), otherwise the crown is very well preserved. The crown has a subconical shape, being mediolaterally compressed, poorly curved lingually and with a slight distal curvature. The labial surface lacks both apicobasal facets (see *Young & Andrade, 2009*; *Andrade et al., 2010*; *Foffa et al., 2018a*; *Herrera, Fernández & Vennari, 2021*) and apicobasal fluting (see *Foffa et al., 2018a*). The root-crown junction is preserved, with the root irregularly broken, with the mesial margin the most complete (Fig. 5A).

Mesial and distal carinae are present in PIN 5477/2451 and are formed by a carinal keel. The keels are continuous from the root-crown junction to the apical region. The carinae are more prominent in the apical half of the mid-crown, but they lack 'carinal flanges' (see *Chiarenza et al., 2015*; *Young et al., 2015a*). Towards the root-crown junction, the carinae become less pronounced, but they remain distinguishable from the surrounding enamel ornamentation (Figs. 5C, 5D). There are very poorly defined, microscopic denticles along the carinae that are difficult to observe even with scanning electron microscopy (*i.e.,* incipient microdenticles, *sensu Young et al., 2013a*) (Figs. 5G, 5H). The denticles are not contiguous along the carinae, *e.g.,* in Fig. 5H there are widely separated microdenticles visible, in short rows of two-six denticles. The superficial enamel ornamentation does not contact the keel (= no false serrations).

Both the labial and lingual surfaces are poorly ornamented, with the ornamentation more pronounced on the latter. The labial surface is largely 'smooth' macroscopically, with short apicobasally aligned ridges on the basal and mid-crown regions (Fig. 5A). The ridges are of low relief, short, and are widely spaced. The lingual surface has a similar morphology, except that there are more apicobasal ridges and they are of higher relief, and they cover approximately 80% of the lingual enamel (Fig. 5B). Apically, the crown becomes macroscopically 'smooth'. The 'smooth' regions of the crown have microscopic poorly defined crests that create a 'wavy'-like pattern (see Fig. 5F).

The enamel ornamentation and denticle morphologies match those of *Tyrannoneustes lythrodectikos* from the UK, and tooth crowns referred to *Tyrannoneustes* sp. from France and Poland (*Young et al., 2013a*; *Foffa & Young, 2014*). Amongst Callovian

metriorhynchids, incipient microdenticles are known for both *Tyrannoneustes* and '*Metriorhynchus*' *brachyrhynchus*, however, they are better defined in the latter (*Young et al., 2013a*). PIN 5477/2451 also differs from '*Metriorhynchus*' *brachyrhynchus* in the lack of apicobasal fluting on the labial surface, the crown is not laminar in cross-section, and the carinae are not as prominent (*Young et al., 2013a*; *Foffa & Young, 2014*; *Foffa et al., 2018a*). Although the lingual surface of PIN 5477/2451 has numerous high-relief enamel ridges that have a higher-relief than those of the labial surface, they do not match those seen in '*Metriorhynchus*' *brachyrhynchus* (see figure 22G in *Young et al., 2013a*).

**Specimen**—PIN 5818/9, isolated tooth crown. Specimen collected in 1928 by P.A. Gerasimov.
**Locality**—Gzhel, near Rechitsy Village, Ramenskoe District, Moscow Oblast, Russia (for details see *Gerasimov et al., 1996*).
**Horizon**—Upper middle Callovian (likely *Erymnoceras coronatum* ammonite Biozone), Middle Jurassic, Kriusha Formation.
**Description**—The tooth crown is largely incomplete, with the apex broken and most of the basal-mid region missing or broken (Figs. 5I–5L). Large regions of the enamel on the labial and lingual surfaces are missing, and the carinae are damaged, particularly apically (Figs. 5I–5M). From what is preserved, the crown has a subconical shape, being mediolaterally compressed, and slightly curved lingually. The labial surface lacks both apicobasal facets (see *Young & Andrade, 2009*; *Andrade et al., 2010*; *Foffa et al., 2018a*; *Herrera, Fernández & Vennari, 2021*) and apicobasal fluting (see *Foffa et al., 2018a*).

This tooth crown (PIN 5818/9) has mesial and distal carinae that is formed by a carinal keel. The keels are continuous in what is preserved of the crown and lack 'carinal flanges' (see *Chiarenza et al., 2015*; *Young et al., 2015a*). The carinae do not become more prominent or more reduced in any specific region, although too much of the crown is missing to determine whether the morphology is consistent along the entire tooth (Figs. 5K, 5L). There are very poorly defined microscopic denticles along the carinae that can only be clearly observed with the use of scanning electron microscopy (*i.e.,* incipient microdenticles, *sensu Young et al., 2013a*) (Fig. 5N). The denticles are not contiguous along what is preserved on the carinae, *e.g.,* in Fig. 5N there are five widely separated microdenticles visible on the carina but apically there is no further evidence of denticles. There are no false serrations (= superficial enamel ornamentation contacting the keel).

Both the labial and lingual surfaces are poorly ornamented. In the preserved basal region, there are apicobasally aligned ridges that are (sub)-parallel to one another. The enamel ridges are of low relief and comparatively short—*i.e.,* they do not continue along most of what is preserved of the crown. The ridges are fewer in number on the labial surface (Fig. 5I) compared to the lingual surface (Fig. 5J), although this could be preservational as more of the lingual surface is preserved. In the apical half of the preserved crown, the enamel is macroscopically 'smooth', although there appear to be microscopic poorly defined crests (best seen in lingual view, Fig. 5J).

As with the previously described tooth crown (PIN 5477/2451), this crown (PIN 5818/9) also matches the UK, French and Polish *Tyrannoneustes* tooth crowns (*Young et al., 2013a*;

*Foffa & Young, 2014*) in its enamel ornamentation and denticle morphologies. As stated above, for Callovian metriorhynchids, incipient microdenticles have only been described for *Tyrannoneustes* and '*Metriorhynchus*' *brachyrhynchus*, with the denticles being better defined in the latter (*Young et al., 2013a*). PIN 5818/9 also differs from '*Metriorhynchus*' *brachyrhynchus* in the lack of apicobasal fluting on the labial surface, the crown is not laminar in cross-section, and not having numerous high-relief enamel ridges on the lingual surface (*Young et al., 2013a*; *Foffa & Young, 2014*; *Foffa et al., 2018a*).

GEOSAURINAE *Bonaparte, 1845* (sensu *Young & Andrade, 2009*)
GEOSAURINI INDET. Morphotype 1 (Fig. 6; Fig. S8)

**Specimen**—SSU 14/31 (present-day catalogue number; SSU 104a/29 in *Arkhangelsky 1999*), tooth crown. Specimen collected in 1984 by M.S. Arkhangelsky.
**Locality**—sovkhoz Leninskiy put' near CHP 5 power station, Saratov, Saratov Oblast, Russia. For more details see *Gulyaev & Ippolitov (2021)*, who described the nearby section 'TETs-5', although the lower part of the succession, from which SSU 14/31 was collected, is not exposed in the 'TETs-5' quarry.
**Horizon**—lower half of the lower Callovian, *Cadoceras elatmae* or *Cadochamoussetia subpatruus* Ammonite Biozone, Middle Jurassic, Khlebnovka Formation.
**Description**—The tooth crown is incomplete, with the apical tip missing, as well as an unknown amount of the basal region (Fig. 6). From what is preserved, the tooth crown would have been subconical, being both lingually curved and slightly mediolateral compressed. The preserved labial surface lacks apicobasal facets (*Young & Andrade, 2009*; *Andrade et al., 2010*; *Foffa et al., 2018a*; *Herrera, Fernández & Vennari, 2021*) but does have apicobasal fluting (*Foffa et al., 2018a*). Fluting, a 'furrowed-grooved' pattern can be seen at the center of the labial surface, is known from three different Callovian taxa, '*Metriorhynchus*' *brachyrhynchus*, *Ieldraan melkshamensis* and an undescribed taxon (all geosaurines, see *Foffa et al., 2018a*).

Mesial and distal carinae are present in SSU 14/31 and are formed by a carinal keel. The keels are continuous in what is preserved of the crown and lack 'carinal flanges' (see *Chiarenza et al., 2015*; *Young et al., 2015a*). The carinae do not become more prominent in any specific region, although in the more apical region the carinae are 'thicker' (Fig. 6C). There are well-defined microscopic denticles along the carinae (*i.e.*, microdenticles, *sensu Andrade et al., 2010*) (Figs. 6G–6J). The denticles are contiguous along the carinae and appear to vary in size and shape, becoming somewhat larger in the more apical region (see Figs. 6G–6H). The superficial enamel ornamentation does not come into contact with the keel (= no false serrations). Interestingly, the carina splits near the base from one side (Figs. 6C, 6G, 6H). Such abnormalities (= supernumerary carinae) are known for some archosaurs and other vertebrate lineages with cutting edges on teeth (*e.g.*, *Erickson, 1995*; *Beatty & Heckert, 2009*; *Itano, 2013*; *Welsh, Boyd & Spearing, 2020*); however, these have never been described for metriorhynchids before.

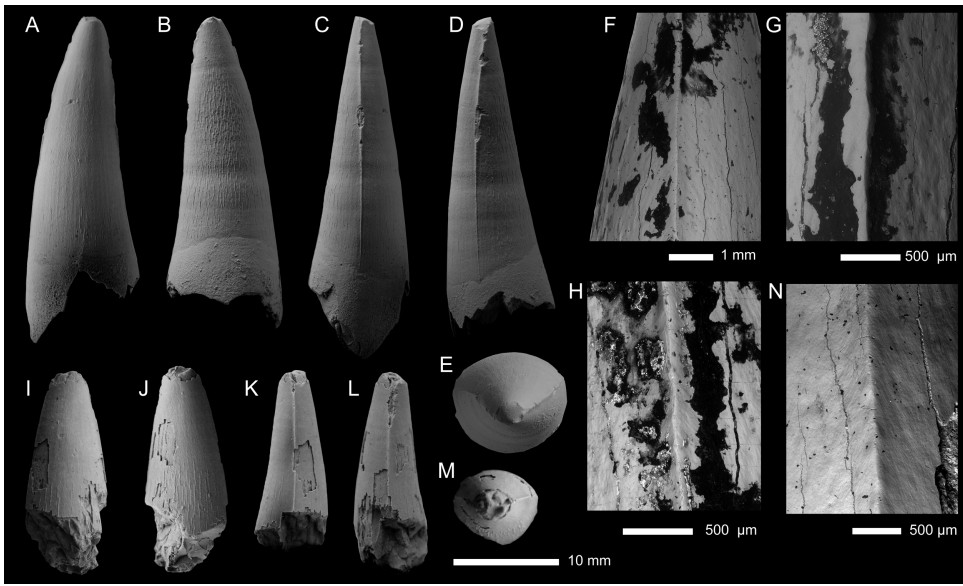

**Figure 5  Metriorhynchid tooth crowns from the middle Callovian, of Moscow and Ryazan oblasts, Russia.** (A–H) *Tyrannoneustes* sp. (PIN 5477/2451); (I–N) *Tyrannoneustes* sp. (PIN 5818/9). Crowns are depicted in labial (A, I), lingual (B, J), mesial (C, K), distal (D, L), and apical (E, M) views. (F–H) and N are SEM photographs of carinae.

Based on what is preserved of the tooth crown, the enamel ornamentation on the labial and lingual surfaces differs. On the labial surface, there are few apicobasally aligned ridges, and when present, are best seen closer to the carinae (Fig. 6C). Instead, there is a pebble-like ornamentation pattern that is more conspicuous in the apical region and begins to disappear more basally (Fig. 6A). On the lingual surface there are some apicobasal ridges, which are discontinuous and well separated from one another (Fig. 5C).

The specimen (SSU 14/31) differs from *Ieldraan melkshamensis* in: (i) the subcircular cross-section of the crown and the lack of apicobasal facets (Figs. 6A, 6C–6F). While it is possible that faceting was present in the basal regions of the crown, based on what is preserved, there is no clear evidence of any distinctive facets. (ii) the carinal keels of SSU 14/31 are not as prominent as in *Ieldraan melkshamensis* (see *Foffa et al., 2018a*: Fig. 4). (iii) the denticles of SSU 14/31 are contiguous and well-developed, whereas in *Ieldraan melkshamensis* the denticles are not completely contiguous and are 'incipient'/poorly developed (*Foffa et al., 2018a*). While it is possible that there is variation along the tooth row in *Ieldraan melkshamensis*, no known metriorhynchid has been found to have both faceted and unfaceted teeth, and no metriorhynchid has some crowns with incipient microdenticulated carinae and  other with well-developed contiguous microdenticulated carinae (*e.g.*, *Andrade et al., 2010*; *Young et al., 2013a*; *Foffa et al., 2018a*). However, the carinae of SGM BX-12, described below, is the first known metriorhynchid tooth crown to display both incipient microdenticles on the same carina as well-developed contiguous denticles (Fig. 7). The contiguous and well-developed denticles also clearly differentiate

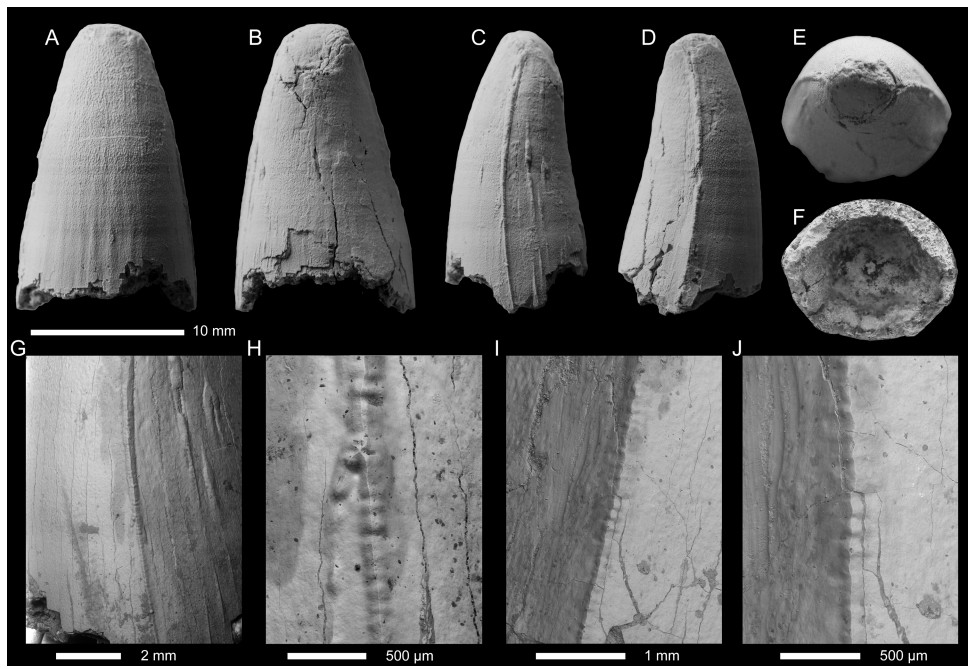

**Figure 6** **Geosaurini indet tooth crown (SSU 14/31) from the lower Callovian of Saratov, Saratov Oblast.** (A–F) Tooth crown in labial (A), lingual (B), mesial and distal (C, D), apical (E), and basal (F) views. (G–J) SEM photographs of the carina.

SSU 14/31 from '*Metriorhynchus*' *brachyrhynchus* and indeterminate geosaurine from (*Foffa et al., 2018a*) (PETMG R248).

This specimen is the oldest known metriorhynchid tooth crown that has denticles that are contiguous along the carinae and are well-defined. Such a combination is widespread amongst Late Jurassic and Early Cretaceous geosaurin metriorhynchids (*e.g.*, *Geosaurus*, *Dakosaurus*, and *Plesiosuchus*) but is unknown for Middle Jurassic taxa *Andrade et al., 2010*; *Young et al., 2013a*; *Foffa et al., 2018a*). The best-sampled dentition of Middle Jurassic geosaurin metriorhynchids comes from the middle and late Callovian of the Oxford Clay Formation of the UK. There, four named species had some form of serrated dentition: '*Metriorhynchus*' *brachyrhynchus*, *Tyrannoneustes lythrodectikos*, *Suchodus durobrivensis* and *Ieldraan melkshamensis*, as well as a specimen closely related to *Dakosaurus* (the 'Mr Leeds dakosaur' = NHMUK PV R 3321) and the indeterminate geosaurine figured by *Foffa et al. (2018a)* (PETMG R248). These species and specimens had incipient microdenticles (*i.e.,* very poorly defined denticles that are microscopic that can only be well observed with the use of scanning electron microscopy) that do not form a contiguous series along the carinae, instead forming short rows denticles (*Young et al., 2013a*; *Foffa et al., 2018a*). We conclude that SSU 14/31 is most likely an unknown species of geosaurin metriorhynchid.

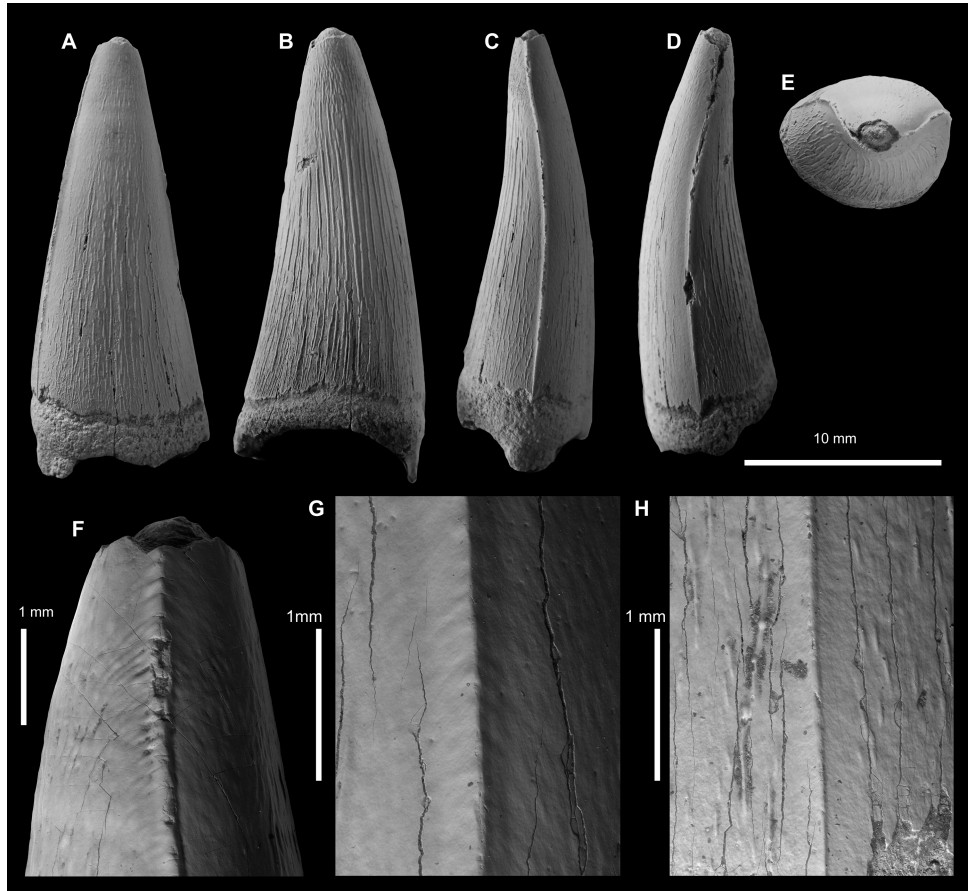

**Figure 7 Geosaurini indet tooth crown(SGM BX-12) from the lower Callovian of Makariev District, Kostroma Oblast.** (A–E) Tooth crown in labial (A), lingual (B), mesial (C), distal (D), and apical (E) views. (F)–(H), SEM photographs of the carina at the apical (F), central (G) and basal (H) parts of the crown.

GEOSAURINI INDET. Morphotype 2 (Fig. 7; Fig. S9)

**Specimen—** SGM BX-12, tooth crown. Specimen collected by A.V. Stupachenko.
**Locality—** Mikhalenino village, Makariev District, Kostroma Oblast, Russia (for details see *Mitta, 2000*).
**Horizon—** *Cadoceras elatmae* ammonite Biozone; lower Callovian, Middle Jurassic, Kologriv Formation.
**Description—**The tooth crown is largely complete, although the apex is broken showing the underlying dentine (Figs. 7A–7F). There is damage to the enamel along the distal carina, and on the lingual surface. The crown is subconical, mediolaterally compressed, and curved both distally and lingually. The labial surface lacks both apicobasal facets (see *Young & Andrade, 2009*; *Andrade et al., 2010*; *Foffa et al., 2018a*; *Herrera, Fernández & Vennari, 2021*) and apicobasal fluting (see *Foffa et al., 2018a*).

 

Mesial and distal carinae are present in SGM BX-12 and are formed by a carinal keel. The keels are continuous and 'carinal flanges' (see *Chiarenza et al., 2015*; *Young et al., 2015a*). The carinae become less prominent near the crown-root junction (Fig. 7A). The serration morphologies along the carinae are highly unusual and have never been observed in a metriorhynchid before (Figs. 7F–7H). Apically, there are denticles that are microscopic, well-defined, and are contiguous (*i.e.,* microziphodont, *Andrade et al., 2010*; *Young et al., 2013a*; *Foffa et al., 2018a*). On the labial surface, some of the superficial enamel ridges interact with the carina, and the denticles, forming the false serration morphology (*Andrade et al., 2010*; *Young et al., 2013a*; Fig. 7F). However, moving basally, the denticles become less well-defined and no longer form a contiguous series (*i.e.,* incipient microziphodonty, *Young et al., 2013a*; *Foffa et al., 2018a*). Even more interesting, is that the denticles disappear in the basal region of the crown (Fig. 7H).

Both the labial and lingual surfaces are well ornamented (Figs. 7A–7E). Apically on the labial surface, the ornamentation becomes less pronounced in terms of relief, although still present (Figs. 7D, 7F). This ornamentation is composed of ridges that are short, irregularly arranged, and sometimes contact the carinae. On the lingual surface, these short, tightly packed, and irregularly arranged ridges are more noticeable (Figs. 7B–7C). In the mid-crown and basal regions, both surfaces have numerous, elongate apicobasally aligned ridges. The ridges are discontinuous and show great variability in length, particularly those closer to the carinae, where the ridges can become exceptionally short (Fig. 7C). On both surfaces, ridges are not present adjacent to the carinae at the mid-crown (Figs. 7C–7E).

Given the presence of well-defined denticles, we can refer this tooth crown to Geosaurini. Only three lineages of Geosaurini are known to have well-defined denticles, Geosaurina, Plesiosuchina, and *Dakosaurus* (*Andrade et al., 2010*; *Young et al., 2013a*; *Chiarenza et al., 2015*; *Foffa et al., 2018a*), and we cannot refer this tooth crown to any of those clades. We cannot refer SGM BX-12 to Geosaurina as it lacks a laminar in cross-section, and lacks apicobasal facets, and 'smooth' enamel ornamentation (*Young & Andrade, 2009*; *Andrade et al., 2010*; *Foffa et al., 2018a*). We cannot refer SGM BX-12 to Plesiosuchina as it lacks 'carinal flanges', a pronounced distal curvature, and sub-rectangular denticles (*Young et al., 2012a*; *Young et al., 2012b*; *Young et al., 2014a*; *Chiarenza et al., 2015*). Finally, we cannot refer it to *Dakosaurus* as the crown lacks the characteristic 'robust' *Dakosaurus* morphology, 'carinal flanges', carinal macrowear facets (although these only appear on fully erupted crowns that have experienced tooth-tooth contact), and 'smooth' enamel ornamentation. Moreover, the carinae of SGM BX-12 transition from true, well-defined denticles apically to incipient denticles in the mid-crown to being unserrated basally. This transition has never been observed before in Thalattosuchia. As such, we can only refer to it as Geosaurini indet.

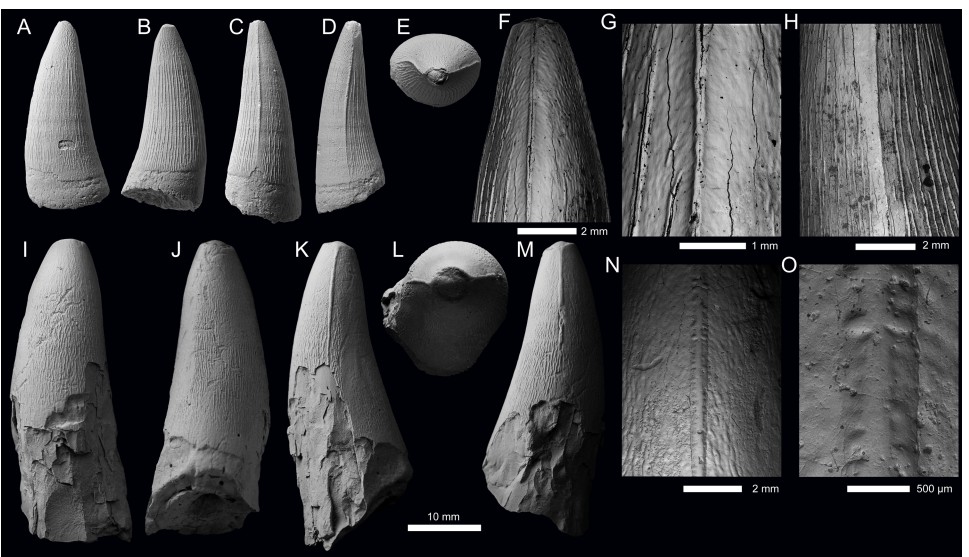

**Figure 8** **Metriorhynchid tooth crowns from the Oxfordian-Kimmeridgian strata.** (A–H) Metriorhynchidae cf. 'E'-clade (PIN 5477/3579) from Rybaki village, Moscow Oblast, Russia. (I–O) *Torvoneustes* sp. (UPM 3026) from Okshovo village, Vladimir Oblast, Russia. Crowns are depicted in labial (A, I), lingual (B, J), mesial (C, M), distal (D, K), and apical (E, L) views. (F–H) and (N–O) are SEM photographs of carinae.

GEOSAURINAE *Bonaparte, 1845* (sensu *Young & Andrade, 2009*)
METRIORHYNCHIDAE cf. 'E'-CLADE (Figs. 8A–8H; Fig. S10)

**Specimen**—PIN 5477/3579, tooth crown. Specimen collected by G.V. Mirantsev in 2009.
**Locality**—Rybaki Village, Ramenskoe District, Moscow Oblast, Russia (for details see *Rogov, 2017*, *Bragin, Bragina & Mironenko, 2023*).
**Horizon**— Found *ex situ*. Upper Oxfordian or lowermost Kimmeridgian, Upper Jurassic, Podmoskov'e, Kolomenskoe or Makaryev Formation. The uppermost part of the section is represented by middle and upper Volgian sands; however, due to the color and type of preservation, the age of the specimen is definitely late Oxfordian–early Kimmeridgian, not Volgian.
**Description**—The tooth crown is largely complete, with only the tip of the apex worn (Figs. 8A–8F). There is some minor damage to the enamel on the labial surface, and the carinae are partly worn (Fig. 8B); otherwise the crown is very well preserved. The crown has a subconical shape, being mediolaterally compressed, and curved both lingually and distally. The labial surface lacks both apicobasal facets (see *Young & Andrade, 2009*; *Andrade et al., 2010*; *Foffa et al., 2018a*; *Herrera, Fernández & Vennari, 2021*) and apicobasal fluting (see *Foffa et al., 2018a*).

Mesial and distal carinae are present in PIN 5477/3579 and are formed by a carinal keel. The keels are continuous from the root-crown junction to the apical region. The carinae are at their most prominent at the mid-crown, but they lack 'carinal flanges' (see

*Chiarenza et al., 2015*; *Young et al., 2015a*). Towards the root-crown junction, the carinae become less pronounced until they look similar to the surrounding enamel ornamentation (Fig. 8C). There are no identifiable denticles along the carinae, although in places the keel becomes irregular, and there could be incipient microdenticles (*Young et al., 2013a*) (Fig. 8G). Apically, some of the superficial enamel ridges abut the keel, but there is no evidence of false serrations.

Both the labial and lingual surfaces are well ornamented. From the base, to approximately 75% of the way towards the apex, there are numerous apicobasally aligned ridges that are (sub)-parallel to one another. These ridges are not as tightly packed as those seen in *Torvoneustes* (*e.g.*, *Andrade et al., 2010*; *Young et al., 2013a*; *Barrientos-Lara et al., 2016*). The ridges are more elongated on the lingual surface than on the labial surface (Figs. 8A–8B). The ridges are less regular on the labial surface, being variable in length and position (Fig. 8B). In this basal-mid crown region, few enamel ridges are close to the carinae, and only very close to the root/crown junction. Apically, there is a shift in ornamentation pattern on both surfaces. Here the ridges become much shorter and much more tightly packed, but do not form an anastomosed pattern like in *Torvoneustes* (*Andrade et al., 2010*; *Young et al., 2013a*; *Barrientos-Lara et al., 2016*). On the labial surface, many of these ridges become exceptionally small (Figs. 8E–8F). At the apex, some of these tiny ridges begin to get closer to the carinae, with some touching the keel (Fig. 8F).

The specimen can be excluded from *Geosaurus* (and Geosaurina) as the crown is not laminar in cross-section, and lacks apicobasal facets, 'smooth' enamel ornamentation, and well-defined contiguous denticles along the carinae (*Young & Andrade, 2009*; *Andrade et al., 2010*; *Foffa et al., 2018a*). It can also be excluded from *Torvoneustes*, as the carinal keels are not prominent and thick, the crown is not as conical as in that genus, the lack of a true anastomosed enamel ornamentation in the apical region and well-defined contiguous denticles along the carinae (*Andrade et al., 2010*; *Young et al., 2013a*; *Young et al., 2013b*; *Young et al., 2020a*; *Barrientos-Lara et al., 2016*). *Dakosaurus* can be excluded as the crown does not have the 'robust' morphology associated with the genus *Dakosaurus*, and lacks 'carinal flanges', carinal macrowear facets (although these only appear on fully erupted crowns that have experienced tooth-tooth contact), 'smooth' enamel ornamentation, and well-defined, macroscopic denticles that are contiguous along the carinae (*Young et al., 2012a*; *Young et al., 2012b*; *Young et al., 2015a*). *Plesiosuchus* (and Plesiosuchina) can be excluded as the crown lacks 'carinal flanges', a pronounced distal curvature, and well-defined, sub-rectangular denticles that are contiguous along the carinae (*Young et al., 2012a*; *Young et al., 2012b*; *Young et al., 2014a*; *Chiarenza et al., 2015*). Late Jurassic rhacheosaurins do not have dentition as diagnostic as those of geosaurins, however their tooth crowns tend to be small, with little to no ornamentation, and those which do have ornamentation have apicobasally aligned ridges with no apical shift in pattern (*e.g.*, *Herrera, Gasparini & Fernández, 2013*; *Herrera, Fernández & Vennari, 2021*; *Sachs et al., 2019*; *Sachs et al., 2021*).

The closest match for PIN 5477/3579 is the 'E'-clade (*sensu Abel, Sachs & Young, 2020*). The specimens referred to this group are still very poorly understood (hence why the genus has not been named). Specimens that pertain to this clade are known from the

late Oxfordian/early Kimmeridgian to early Tithonian of England, France, Germany, and Switzerland (see *Abel, Sachs & Young, 2020*; *Young et al., 2020b*). Tooth crowns of these taxa range from being mediolaterally compressed to almost as robust as *Dakosaurus* teeth (*Lepage et al., 2008*; *Abel, Sachs & Young, 2020*); the 'Passmore crocodile' –OUMNH J1583). The distribution of serrations in this clade is not well understood, but the specimen from Germany described by *Abel, Sachs & Young (2020)* had irregular microscopic denticles (incipient microziphodonty, *Young et al., 2013a*), while it is unclear if all the teeth of the 'Passmore crocodile' (OUMNH J1583) have denticles. The 'E'-clade tooth crowns undergo a dramatic change in enamel ornamentation near the apex (but not the same as that seen in *Torvoneustes*). In the basal-mid crown region, there are elongate apicobasal ridges, while close to the apex there is a shift to much shorter and more closely packed ridges (Figs. 8A–8F; *Abel, Sachs & Young, 2020*). Given that the enamel ornamentation of 'E'-clade specimens is the most similar to PIN 5477/3579, and that it lacks the apomorphies of other known lineages, we refer to it as Metriorhynchidae cf. 'E'-clade.

GEOSAURINAE *Bonaparte, 1845* (sensu *Young & Andrade, 2009*)
TORVONEUSTES sp. (Figs. 8I–8O; Fig. S11)

**Specimen**—UPM 3026, tooth crown. Specimen collected by A.P. Ippolitov in 2020.
**Locality**—Okshovo village, Melenki district, Vladimir Oblast, Russia (details of the local geology see in *Ippolitov & Shchepetova, 2020*).
**Horizon**— Found *ex situ* on the beach. The preservation suggests that the specimen derived from the *Cardioceras densiplicatum* ammonite Biozone, middle Oxfordian, Upper Jurassic, Unzha Formation—the only Jurassic unit available above the water level. At the same locality, Hauterivian fossils occur intermixed with Oxfordian invertebrates (Ippolitov, unpubl.); however, fossils of Hauterivian age have recognizable preservation –eroded with a surface covered by oolite marl.
**Description**—The morphology of UPM 3026 (Figs. 8I–8O) is consistent with the dental morphology seen in the genus *Torvoneustes* from the Kimmeridgian–Tithonian of the UK and Mexico (see *Andrade et al., 2010*; *Young et al., 2013a*; *Young et al., 2013b*; *Young et al., 2020a*; *Barrientos-Lara et al., 2016*), and possibly also from the Valanginian of the Czech Republic (*Madzia et al., 2021*). The tooth crown is incomplete, with the apex missing due to apical wear, and the basal region very poorly preserved (Figs. 8I–8M). Much of the enamel is broken in the basal-lingual region, with then the underlying dentine damaged (Figs. 8I, 8K, 8M). The broken (or worn) apex is somewhat rounded and blunt (Figs. 8J–8M). The tooth crown has a conical shape, being poorly mediolaterally compressed, particularly basally. It is lingually curved, more so in the 'mid' region of the crown. The labial surface lacks both apicobasal facets (see *Young & Andrade, 2009*; *Andrade et al., 2010*; *Foffa et al., 2018a*; *Herrera, Fernández & Vennari, 2021*) and apicobasal fluting (see *Foffa et al., 2018a*).

Mesial and distal carinae are present in UPM 3026 and are formed by a carinal keel. The keels are very prominent and thick, an autapomorphic feature of the genus *Torvoneustes* (*Andrade et al., 2010*; *Young et al., 2013a*; *Young et al., 2013b*; *Young et al., 2020a*) and is

present in an Early Cretaceous tooth crown possibly referrable to *Torvoneustes* (*Madzia et al., 2021*). Although prominent, they lack the 'carinal flanges' seen in *Dakosaurus* and members of Plesiosuchina (see *Chiarenza et al., 2015*; *Young et al., 2015a*). There are serrations present along the keel (Figs. 8N, 8O). The carinae and denticles are similar to those described for *Torvoneustes carpenteri* (*Andrade et al., 2010*; *Young et al., 2013a*), with the carinae being homogenous, having a long series of contiguous true denticles that are microscopic. It is unclear whether they were poorly defined in life or their morphology is the result of postmortem taphonomic processes. The denticles typically do not exceed 300 μm in dimensions (microziphodont condition; *Andrade et al., 2010*). The enamel ornamentation does approach the carinae and seems to abut the denticles (Figs. 8M, 8N), but we do not see any instances of the superficial enamel ornamentation continuing onto the keel itself. Therefore, this tooth crown does not exhibit a false ziphodont morphology, making it similar to *Torvoneustes coryphaeus* (*Young et al., 2013b*), and differing from *To. carpenteri*, *To. mexicanus* and *Torvoneustes* sp. (?early Tithonian tooth crowns), all of which have false serrations (*Andrade et al., 2010*; *Young et al., 2013b*; *Young et al., 2020a*; *Barrientos-Lara et al., 2016*). The presence/absence of false serrations in the genus *Torvoneustes* could have taxonomic implications, or it could be the result of natural variation across the tooth-row, unfortunately there are no specimens complete enough to make such a determination.

The enamel on the labial and lingual surfaces is heavily ornamented. In the basal-and-mid regions of the crown, the ornamentation is composed of numerous, tightly packed, apicobasally aligned ridges that are arranged (sub)-parallel to one another. The ridges are more elongated on the lingual surface than on the labial surface (Figs. 8I–8J). Towards what is preserved of the apex, the enamel ornamentation shifts, going from a tightly packed ridged pattern to a less defined and more anastomosed one (Figs. 8L, 8M). The superficial enamel ornamentation is clearly separate from the carinae in the basal-mid crown regions (Fig. 8K), but closer to the apex, the ornamentation begins to abut the carinae (Figs. 8M, 8O). These shifts in enamel ornamentation pattern occur in all known *Torvoneustes* tooth crowns (*Andrade et al., 2010*; *Young et al., 2013a*; *Young et al., 2013b*; *Young et al., 2020a*; *Barrientos-Lara et al., 2016*).

The dentition of *Torvoneustes* was similar to those seen in the teleosauroid subclade Machimosaurini (particularly the genus *Machimosaurus*). Both thalattosuchian lineages shared tooth crowns that were poorly curved lingually, had a blunt apex, serrated carinae, and a characteristic enamel ornamentation composed of an anastomosed pattern apically, and numerous apicobasally aligned ridges in the basal-and-mid crown regions. The most geographically widespread machimosaurin was *Machimosaurus hugii*, however this species has 'pseudodenticles' along the superficial enamel ridges, a morphology only found in *M. hugii* and *M. rex* (*Young et al., 2014b*; *Fanti et al., 2016*). Moreover, *Machimosaurus* species have variable carinae morphologies, with tooth crowns either lacking carinae or having low carinae (*Young et al., 2014b*; *Young et al., 2014c*). UPM 3026 lacks 'pseudodenticles' and the carinae are very prominent, the latter being characteristic of *Torvoneustes* (*Young et al., 2020a*).

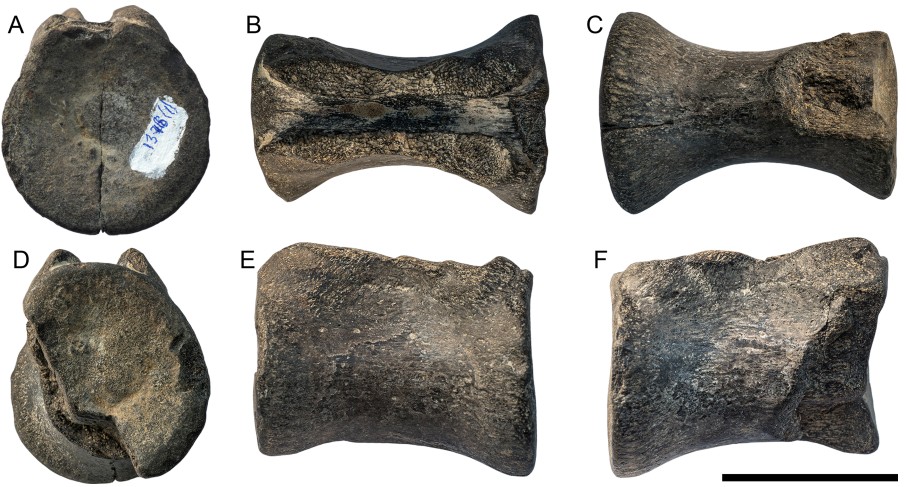

**Figure 9  Large dorsal vertebra (Thalattosuchia indet. , UPM 1376) from the Oxfordian of Tarkhanovskaya pristan, Republic of Tatarstan, Russia; in anterior.** (A) Dorsal, (B) ventral, (C) posterior, (D) and lateral (E, F) views. Scale bar represents five cm. Photo credit: Dmitry Grigoriev.

THALATTOSUCHIA INDET. (Fig. 9)

**Specimen**—UPM 1376, an incomplete dorsal vertebra. Specimen collected by D.V. Efimov (*Efimov, 2010*).
**Locality**—Tarkhanovskaya pristan', Tetyushi district, Republic of Tatarstan, Russia (for details see *Mitta, 2003*).
**Horizon**—lower Oxfordian, Late Jurassic, Vechkusy Formation.
**Description**—Only the dorsal vertebral centrum is preserved. It is approximately eight cm long anteroposteriorly. The articulation surface for the centrum-neural arch is visible, suggesting that this suture was not fused in life. If so, then this specimen would not have come from a morphologically mature individual. The vertebra closely resembles those of other thalattosuchians (*e.g.*, *Fraas, 1902*; *Arthaber, 1906*; *Andrews, 1913*), with an hourglass shape in ventral aspect (Fig. 9C). The shape of the centra in anterior/posterior aspect is sub-circular (Figs. 9A, 9D), and both are poorly concave. Given the preservation, we cannot ascertain which thalattosuchian subclade this specimen pertains to.

METRIORHYNCHIDAE INDET. (Fig. 10)

**Specimens**—UPM 3024, cervical vertebra, collected by I.M. Stenshin; UPM 3031, dorsal vertebra collected by Sergey I. Buganin (UPM); UPM 3025, caudal vertebra collected by Maxim S. Pichugin (UPM); UPM 3030, caudal vertebra collected by Arseny Grishin. Specimens collected in 2021 and 2022.
**Locality**—Gorodischi, Ulyanovsk district, Ulyanovsk Oblast, Russia (for details see *Rogov, 2010*; *Rogov, 2013*).

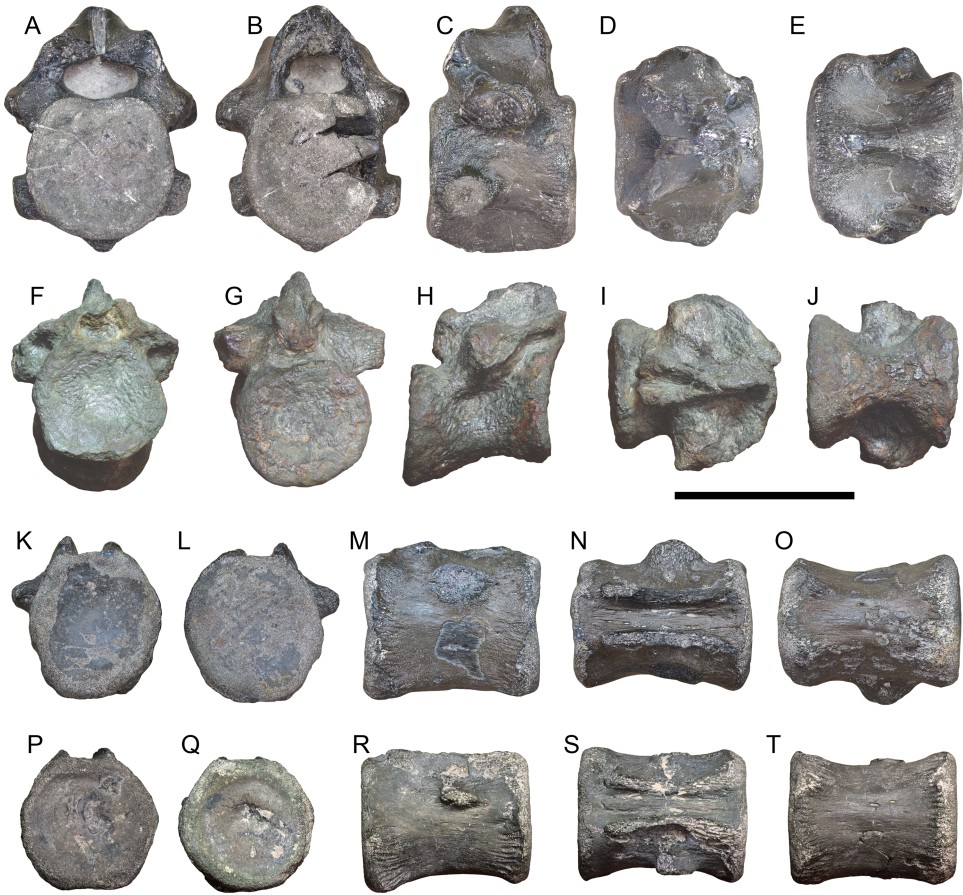

**Figure 10 Vertebrae from the Volgian of Gorodischi locality in Ulyanovsk Oblast, Russia.** (A–E) Cervical vertebra (UPM 3024); (F–J) dorsal vertebra (UPM 3031); (K)–(O), caudal vertebra (UPM 3025); (P)–(T), caudal vertebra (UPM 3030). Vertebrae are depicted in anterior (A, F, K, P), posterior (B, G, L, Q), lateral (C, H, M, R), dorsal (D, I, N, S), and ventral (E, J, O, T) views. Scale bar represents five cm.

**Horizon**—Collected *ex situ* on the beach. The preservation suggests that specimens most likely originate from either the lower Volgian strata (UPM 3024, 3025, 3030) or middle Volgian *Dorsoplanites panderi* ammonite Biozone (strongly pyritized UPM 3031), Upper Jurassic. Trazovo to Promza Formations (*sensu Rogov, 2021b*).

**Description**—The vertebra (UPM 3024) is from the cervical column, as the parapophyses are born entirely on the centrum (*Andrews, 1913*). As the parapophyses are located low on the centrum, it cannot be the fifth post-axial cervical (as the parapophyses would be located in a more dorsal position on the centrum, *e.g.*, *Andrews, 1913*; *Wilkinson, Young & Benton, 2008*). The parapophyses are short, and distally have an articulation surface for their attachment to the capitulum of the cervical ribs. The diapophyses are located on the neural arch, and similarly short, and are oriented ventrolaterally (Figs. 10A, 10C). The centrum is wider than long, which is only known to occur in metriorhynchine metriorhynchids, although this character is very poorly sampled for Late Jurassic and Early Cretaceous geosaurine metriorhynchids (a phylogenetic character from *Young et al., 2021*).

The neural arch is incomplete, with the neural spine and zygapophyses missing. The neural canal is noticeably wider than tall. Curiously, the specimen has a well-developed hypapophysis, extending ventrally below the centrum, and is visible in both anterior and posterior views (Figs. 10A, 10B). Thalattosuchian hypapophyses are generally very poorly developed, being mediolaterally thick (*i.e.,* not forming a laminar blade) and follow the contour on the centrum if present (*e.g.*, see the figures in *Andrews, 1913*; *Young et al., 2013a*; *Sachs et al., 2019*; *Sachs, Young & Hornung, 2020*; *Sachs et al., 2021*). Some thalattosuchian cervical vertebrae have a hypapophyseal keel that forms a largely horizontal shape in lateral view (rather than following the contour on the centrum and being concave); when this happens, they tend to occur on the first, and sometimes also the second, post-axial cervical vertebrae (*e.g.*, NHMUK PV R 9731). This specimen (UPM 3024) has a distinctly convex hypapophysis in lateral view, such that it clearly extends below the centrum (Fig. 10C). This morphology has not been observed in metriorhynchids before.

The dorsal vertebra (UPM 3024) is highly distorted, with the anterior and posterior aspects of the centrum no longer in the same sagittal plane (Fig. 10H). Moreover, the neural spine is missing, as are most of the transverse processes; the prezygapophyses are missing, while the postzygapophyses are distorted. The vertebral centrum is hourglass-shaped, with concave articular surfaces (Figs. 10H, 10J). The anterior and posterior articular surfaces are largely circular in shape and concave (Figs. 10F, 10G).

Only the centra of the two caudal vertebrae (UPM 3025, UPM 3030) are preserved. Both have slightly sub-hexagonal articular surfaces (Figs. 10K, 10L, 10P, 10Q), allowing us to refer this specimen to Metriorhynchidae (based on the caudal series of *Thalattosuchus superciliosus* GLAHM V990, MB.R.5615; *Gracilineustes leedsi* NHMUK PV R 3014, NHMUK PV R 4762, and '*Metriorhynchus*' *brachyrhynchus* NHMUK PV R 3804), teleosauroids and early-diverging metriorhynchoids had rounded articular faces (see *Andrews, 1913*; *Ősi et al., 2018*). Both vertebrae have the characteristic hourglass shape of metriorhynchid vertebrae. The anterior articular surfaces are concave, while the posterior articular surfaces are more flattened. Given the well-developed transverse processes on both vertebrae, they must come from the proximal-most part of the caudal column (*Fraas, 1902*; *Arthaber, 1906*; *Andrews, 1913*; *Sachs et al., 2021*).

METRIORHYNCHIDAE INDET. (Fig. 11)

**Specimen**—KGM No 44, caudal vertebra. Collector is unknown, probably collected by Yu S. Rubtsov around 1995–2003.
**Locality**—Vyatka-Kama phosphate field, Kirov Oblast, Russia (for details see *Morozov et al., 1967*; *Zverkov et al., 2018*).
**Horizon**— Ryazanian (Berriasian) or Valanginian, Early Cretaceous. Katarzhata Formation.
**Description**—The vertebra is a poorly preserved centrum of a caudal vertebra. Given the sub-hexagonal shape of the preserved articular surface (Fig. 11A), this specimen can be referred to Metriorhynchidae (based on the caudal series of *Thalattosuchus superciliosus*

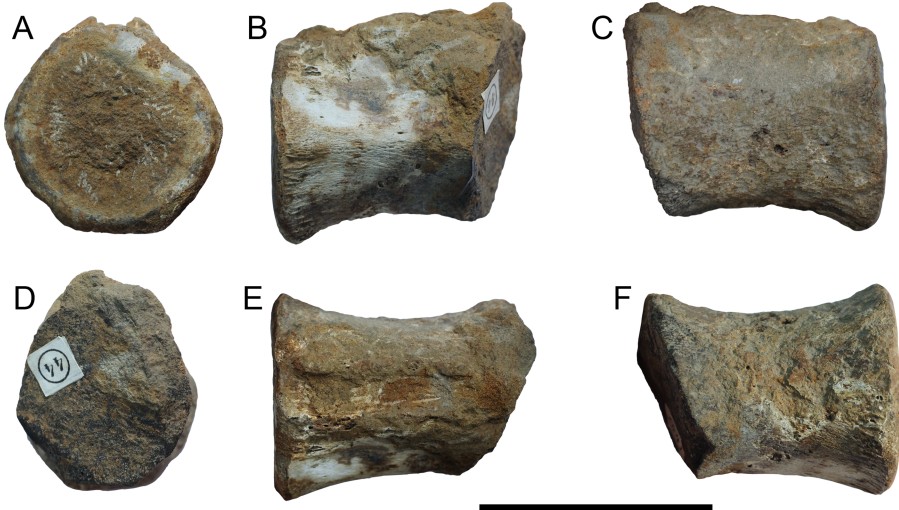

**Figure 11  Caudal vertebra (KGM No 44) from the Ryazanian (Berriasian) or Valanginian of Vyatka-Kama phosphate field, Kirov Oblast, Russia; in anterior (A), lateral (B, C), posterior (D), dorsal (E), and ventral (F) views.** Scale bar represents five cm.

GLAHM V990, MB.R.5615; *Gracilineustes leedsi* NHMUK PV R 3014 and '*Metriorhynchus*' *brachyrhynchus* NHMUK PV R 3804). Although highly incomplete, the vertebra does have an hourglass shape, and the centrum articular face appears to be slightly concave. Given the preservation of the vertebra, it is hard to be sure where in the caudal series it originated from, but the broad articular surface and lack of an extremely compressed morphology suggests it came from the preflexural column (*Fraas, 1902*; *Arthaber, 1906*; *Andrews, 1913*; *Sachs et al., 2021*).

# DISCUSSION

## Taxonomic diversity

The tooth crown from the early Bajocian (PIN 5819/5; Fig. 2) cannot be precisely taxonomically identified. The thalattosuchian fauna of the Aalenian–Bajocian is poorly understood. The diversity of Teleosauroidea is poorly sampled during this time span (*Buffetaut, 1982*; *Johnson, Young & Brusatte, 2020*), while metriorhynchoids were rapidly becoming more aquatically adapted (see *Wilberg, 2015*; *Aiglstorfer, Havlik & Herrera, 2020*; *Cowgill et al., 2022*). As mentioned above, we cannot exclude the possibility that PIN 5819/5 is from an early-diverging metriorhynchoid; however, we have identified it as Thalattosuchia indet.

The morphology of the Middle–Late Jurassic tooth crowns from Russia is consistent with those known from Western Europe. There are two tooth crowns we refer to cf. *Thalattosuchus*. The first, MRUM 1315/1 (Figs. 4A–4F), is from the lower Callovian of the Republic of Mordovia, while the second, PIN 5477/3253 (Figs. 4G–4K), is from the middle Callovian of the neighboring Ryazan Oblast. The three incomplete tooth crowns from the lower Callovian of the Kostroma Oblast could pertain to *Thalattosuchus* (PIN

5819/1, PIN 5819/2, and PIN 5819/6; Figs. 3A–3S), but they are too incomplete for us to identify them as anything but Metriorhynchidae indet. Prior to this paper, the presence of *Thalattosuchus* in the early Callovian was limited to Poitou, France (*e.g.*, *Vignaud, 1995*; *Vignaud, 1997*). The crown MRUM 1315/1 suggests that *Thalattosuchus* had a broad geographical range across Europe in the early Callovian, spanning from France to Russia, and PIN 5477/3253 suggests that the genus remained in the Middle Russian Sea at least until the middle Callovian. The majority of *Thalattosuchus* specimens are known from middle–late Callovian deposits of the UK and late Callovian–early Oxfordian deposits of northern France (*e.g.*, *Eudes-Deslongchamps, 1867–69*; *Andrews, 1913*; *Lepage et al., 2008*; *Young et al., 2021*).

There are two tooth crowns we refer to *Tyrannoneustes* sp.: PIN 5477/2451 from the Ryazan Oblast (Figs. 5A–5H) and PIN 5818/9 from the Moscow Oblast (Figs. 5I–5N). We also cannot preclude that PIN 5819/7 (Figs. 3T–3Y), collected from somewhat the same locality as PIN 5818/9 and approximately from the same bed, is an incomplete *Tyrannoneustes* tooth crown. These teeth are all from the middle Callovian strata, meaning they are the same age as the holotype and the referred specimens of *T. lythrodectikos* from the Oxford Clay Formation of the UK (*Young et al., 2013a*; *Foffa & Young, 2014*). Note, the '*Tyrannoneustes lythrodectikos*' specimen from Germany (*Waskow, Grzegorczyk & Sander, 2018*) lacks all of the autapomorphies of this species and requires re-examination. Other tooth crowns previously referred to *Tyrannoneustes* come from the higher interval, the late Callovian–early Oxfordian of France and Poland (*Young et al., 2013a*). The presence of *Tyrannoneustes* in Russia during the middle Callovian expands the known geographical range of the genus, suggesting it was widespread across the shallow seaways covering much of Europe at that time.

Potentially two more geosaurine species were present in the Middle Russian Sea during the Callovian. The first unusual tooth crown SSU 14/31 (Fig. S2) is poorly preserved, and we cannot make a definitive identification. However, there is evidence of apicobasal fluting on the centre of the labial surface. As mentioned above, this morphology is only known from three metriorhynchid taxa during the Callovian, all of which are geosaurines (*Foffa et al., 2018a*). However, as noted above, SSU 14/31 cannot be referred to any of these three geosaurine taxa ('*Metriorhynchus*' *brachyrhynchus*, *Ieldraan melkshamensis* and an undescribed taxon, see *Foffa et al., 2018a*). Most surprising about this tooth crown is the presence of well-defined and contiguous denticles along the carinae. The presence of such serration morphologies in the early Callovian complicates our understanding of denticle evolution in Metriorhynchidae. *Foffa et al. (2018a)* hypothesised that the contiguous, well-developed serration morphologies seen in the Late Jurassic genera *Dakosaurus*, *Geosaurus*, and *Plesiosuchus* evolved independently three times from Middle Jurassic taxa that had incipient microdenticles. However, SSU 14/31 is the oldest known metriorhynchid with serrated dentition, and it has a morphology very similar to those of Late Jurassic metriorhynchids. It hints that there could be an unknown lineage of Middle Jurassic metriorhynchids that evolved functionally serrated dentition millions of years prior to the better-known taxa of the Late Jurassic. Future discoveries will be needed to test this hypothesis.

The second peculiar tooth crown, SGM BX-12, is perhaps consistent with the 'multiple well-developed serration' hypothesis of *Foffa et al. (2018a)*. Prior to the description of this tooth crown, no known thalattosuchian had dentition with both well-defined denticles and incipient denticles, let alone on the same tooth crown. However, this specimen has both morphologies along the carinae, with one morphology transitioning into another. The specimen SGM BX-12 is, therefore, a key 'transitional' form in metriorhynchid serration evolution. Given that three geosaurin lineages (*e.g.*, Geosaurina, Plesiosuchina, *Dakosaurus*) made the transition from Middle Jurassic species with incipient denticles to Late Jurassic taxa with well-defined denticles that contiguously form a cutting edge, we hypothesise that similar serration morphologies as SGM BX-12 will be found in all three lineages.

Our data shows that during the Callovian, at least three or four metriorhynchid species were present in the Middle Russian Sea and that there probably should be some taxonomic overlap with metriorhynchids known from Western Europe (*e.g.*, *Eudes-Deslongchamps, 1867–69*; *Andrews, 1913*; *Buffetaut, 1982*; *Lepage et al., 2008*; *Young et al., 2013a*, *Foffa et al., 2018a*). However, the two early Callovian geosaurin teeth have morphologies not observed in known material from Western Europe. This illustrates that our understanding of metriorhynchid evolution is being hindered by being too focused on the fossil record of Western Europe, and that there are more surprises to be discovered.

From the Late Jurassic strata of Central Russia, we identify *Torvoneustes* sp. and Metriorhynchidae cf. 'E'-clade (Fig. 6), again, indicating taxonomic overlap between Eastern and Western Europe. *Torvoneustes* sp. (UPM 3026; Figs. 8I–8O) from the middle Oxfordian of the Vladimir Oblast is the second record of the genus from the middle Oxfordian: *Young (2014)* referred an incomplete skull from the UK, that is most likely from the middle Oxfordian to cf. *Torvoneustes*. As mentioned above, the morphology of UPM 3026 has a striking similarity to the *Torvoneustes* tooth crowns from the Kimmeridgian–Tithonian of the UK and Mexico (*Andrade et al., 2010*; *Young et al., 2013a*; *Young et al., 2013b*; *Young et al., 2020a*; *Barrientos-Lara et al., 2016*), and the *Torvoneustes*? sp. tooth crown from the Valanginian of the Czech Republic (*Madzia et al., 2021*). The presence of *Torvoneustes* in Russia expands the known geographical range of this genus to European Russia.

There appears to be a member of the enigmatic 'E'-clade (*Abel, Sachs & Young, 2020*) present in the Middle Russian Sea during the latest Oxfordian or early Kimmeridgian (PIN 5477/3579; Figs. 8A–8H). These geosaurine metriorhynchids remain poorly understood and have not yet been formally named. Specimens referred to this clade are known from across Western Europe during the Late Jurassic, including France, Germany, Switzerland, and the UK (*Lepage et al., 2008*; *Abel, Sachs & Young, 2020*; *Young et al., 2020b*). Their putative presence in Russia expands their known geographical range and suggests that this clade/taxon was widespread during the Late Jurassic. It is unclear whether this taxon was rare, has been overlooked in museum collections, or lived in environments that are not well sampled. In particular, the 'E'-clade appears to have been absent from the lagoonal ecosystem deposits of southern France and southern Germany, where numerous species of other Late Jurassic thalattosuchians have been discovered (*e.g.*, see *Fraas, 1902*; *Young & Andrade, 2009*; *Andrade et al., 2010*; *Young et al., 2012a*; *Young et al., 2012b*; *Sachs et al.,*

*2019*; *Sachs et al., 2021*; *Johnson, Young & Brusatte, 2020*; *Herrera, Aiglstorfer & Bronzati, 2021*). Given that this clade also shows evidence of true posterodorsal retraction of the external nares (*Young et al., 2020b*), it is possible that the 'E'-clade frequented deeper water ecosystems than the better studied metriorhynchids of the Late Jurassic.

Based on the revised tooth crown morphology guilds outlined by *Foffa et al. (2018b)*, thalattosuchians occupied three different guilds in the Middle Russian Sea. The Middle Jurassic tooth crowns fall into the 'pierce' guild, as they do in the Oxford Clay Sea–Kimmeridge Clay Sea (*Foffa et al., 2018b*). During the Late Jurassic, two new guilds are occupied –the 'crunch' guild by *Torvoneustes* and the 'cut' guild by the 'E'-clade tooth taxon. The expansion into these two guilds also occurred in the Oxford Clay Sea–Kimmeridge Clay Sea during the Late Jurassic (*Foffa et al., 2018b*). Given the taxonomic overlap between Eastern and Western Europe, it is possible there was little endemism in open shelf environments across Europe.

The vertebral elements from the Late Jurassic and Early Cretaceous described herein cannot be referred to any metriorhynchid subclade with certainty. The Oxfordian dorsal vertebra (UPM 1376, Fig. 9) can only be referred to Thalattosuchia indet. The Volgian vertebrae (UPM 3024, UPM 3025, UPM 3030, UPM 3031, Fig. 10) can be referred to Metriorhynchidae, as can the Early Cretaceous caudal vertebra (KGM No 44, Fig. 11). Not only do some of the Callovian teeth from Russia display unusual morphologies, so do some of the Volgian vertebrae. The cervical UPM 3024 is the first known metriorhynchid to have a well-developed hypapophysis, while the incomplete skeleton described by *Hua, Vignaud & Efimov (1998)* had caudal neural spines least as tall as the centrum plus the neural arch. This latter character has only been observed in *Cricosaurus bambergensis* (*Sachs et al., 2019*) and in *Thalattosuchus superciliosus* (NHMUK PV R 2054, although note many metriorhynchid specimens from the Oxford Clay Formation have broken caudal neural spines).

The most significant differences in teleosauroid and metriorhynchid endemism so far observed within Europe are between different Late Jurassic depositional environments. The shallow lagoonal environments of southern France and southern Germany have a diverse assemblage of rhacheosaurin metriorhynchids, with fewer species of geosaurine metriorhynchids, machimosaurine machimosaurids and aeolodontin teleosaurids (*e.g.*, *Andrade et al., 2010*; *Young et al., 2012a*; *Young et al., 2012b*; *Johnson, Young & Brusatte, 2020*; *Young & Steel, 2020*; *Herrera, Aiglstorfer & Bronzati, 2021*). Whereas coastal shelf environments had different assemblages, such as one from northern France, which was dominated by machimosaurine machimosaurids, with metriorhynchine and geosaurine metriorhynchids also being present (*Hua, 1999*; *Lepage et al., 2008*; *Young & Steel, 2020*; *Johnson, Young & Brusatte, 2020*). The deeper water, open- and outer-shelf environments of the Kimmeridge Clay Formation of the UK differ again. There, teleosauroids are rare, with a low number of machimosaurine machimosaurid and aeolodontin teleosaurid fossils discovered, metriorhynchine metriorhynchids are uncommon, but there is a great diversity of geosaurine metriorhynchids (*Young et al., 2012a*; *Young et al., 2012b*; *Young et al., 2020b*; *Young et al., 2014a*; *Young et al., 2014b*; *Foffa et al., 2018b*; *Foffa et al., 2019*; *Young & Steel, 2020*; *Johnson, Young & Brusatte, 2020*). The same three tooth crown guilds ('crunch',
'pierce' and 'cut') are occupied in all of these depositional environments, albeit not by the same genus or the same higher clade (*Foffa et al., 2018b*). It is possible that this pattern was replicated across the Middle Russian Sea, however, future discoveries will be necessary to confirm this.

## Palaeolatitudinal distribution of Metriorhynchidae and their tolerance for low temperatures

The majority of extant crocodylian species live entirely within the Tropics (approximately between 23.4° North and South). However, extant species have a full latitudinal range of approximately 36°N to 36°S, with the American alligator (*Alligator mississippiensis*) and the Yacaré caiman (*Caiman yacare*) representing the northernly and southernly extremes, respectively (*Griggs & Kirshner, 2015*: 340). Other species have ranges similarly close to the subtropics–temperate zone boundary, such as the Indian gharial (*Gavialis gangeticus*; 34°N), the Mugger crocodile (*Crocodylus palustris*; 34°N), the Nile crocodile (*Crocodylus niloticus*; 34°S) and the Broad-snouted caiman (*Caiman latirostris*; 34°S) (*Griggs & Kirshner, 2015*; also see the IUCN red list assessments, at https://www.iucnredlist.org). Prior to extensive habitat destruction, the Chinese alligator (*Alligator sinensis*) had a range as far north as 35° (*Thorbjarnarson & Wang, 2010*).

Given the known ranges of extant crocodylians, we can assume that extinct crocodylomorphs that were semi-aquatic and ectothermic had distributions spanning tropical and subtropical climatic zones. However, during much of the Mesozoic, there was little-to-no glaciation, and tropical climatic zones had a broader latitudinal range than today (*e.g.*, *Boucot, Xu & Scotese, 2013*). Therefore, sea surface temperatures would have been greater at higher latitudes than today. This is demonstrated by the Peterborough Member of the Oxford Clay Formation, where an abundance of thalattosuchian fossils have been discovered, which during the middle Callovian was at palaeolatitude 38°N but had an estimated sea surface temperature of 20–27 °C (*Dromart et al., 2003*). It is, therefore, a reasonable assumption that extant species would have had a broader latitudinal distribution had they lived during the Mesozoic and that thalattosuchians had a broader latitudinal range than those of extant species. Based on palaeolatitudinal estimates, at least three previously known metriorhynchid species exceeded the extant crocodylian latitudinal range within the Southern Hemisphere (see Table S1, and using paleolatitude.org):

- *Cricosaurus araucanensis* 33–40° South (based on 140 and 150 Ma)
- *Cricosaurus lithographicus* 35–40° South (based on 140 and 150 Ma)
- *Dakosaurus andiniensis* 32–40° South (based on 140 and 150 Ma)

Additionally, an indeterminate geosaurine from the Middle Jurassic (late Bathonian) of Argentina (*Gasparini, Cichowolski & Lazo, 2005*) is estimated to have been 40–41° South (based on 160 and 170 Ma). As such, very early in their evolution, metriorhynchids reached further south than extant crocodylians are known today.

There was less data regarding Northern Hemisphere prior to this contribution. The indeterminate Scottish teleosauroid described by *Kean et al. (2021)* would have

been approximately 38–44° North (based on 170 and 190 Ma estimates). The early-diverging metriorhynchoids *Pelagosaurus typus* and *Opisuchus meieri* would have reached approximately 37–42° and 32–39° North, respectively, while Western European metriorhynchids are estimated to have an upper latitudinal range of 39° North (see Table S1). Based on the specimens described herein, we can conclude that metriorhynchids found in Russia could frequent palaeolatitudes as far as north as 50° (Table 2). The Callovian specimens are particularly interesting, as they seem to be the first known metriorhynchids probably associated with low-temperature ecosystems. Eight specimens derive from the interval with palaeotemperatures estimated to be 5–13 °C (*Wierzbowski et al., 2020*; see Table 2), with five different taxonomic assignments: Metriorhynchidae indeterminate, cf. *Thalattosuchus*, Geosaurini indeterminate (morphotype 1), Geosaurini indeterminate (morphotype 2), and *Tyrannoneustes* sp. Therefore, at least three metriorhynchid species frequented regions of the Middle Russian Sea that had temperatures of 10−13 °C, and possibly lower, during the early-to-earliest middle Callovian. Note, the isotopic data from *Wierzbowski et al. (2020)* is based mainly on more offshore sections, while the teeth mentioned here come from shallower environments. The temperature could be slightly higher in these nearshore environments.

The Middle Russian Sea was a transitional zone between the Tethys and Boreal climatic belts, with sea temperatures influenced by both colder Arctic and warmer Peri-Tethys waters. This seaway was inhabited by molluscs of Boreal, Subboreal, and Tethyan origins, which had a dynamic distribution, with periodic southerly range expansions of the Boreal fauna and northernly range expansions of the Tethys fauna (*Rogov, Zakharov & Kiselev, 2008*). During both the *Cadoceras elatmae* chron of the early Callovian and the *Kosmoceras jason* chron of the middle Callovian, Boreal molluscs migrated into more southernly latitudes in the Middle Russian Sea and further to the south, reaching 32–33°N (Crimea, Northern Caucasus, and Turkmenistan; *Rogov, Zakharov & Kiselev, 2008*). *Arkell (1956)* called such episodes 'the Boreal spread.' Eight of the metriorhynchid tooth crowns we describe herein coincide by age with two southernly range expansions of Boreal molluscs, with: Geosaurini indet. (morphotype 1) present at 45°N during the *Cadoceras elatmae –Cadochamoussetia subpatruus* chrons of the early Callovian (Fig. 12), cf. *Thalattosuchus* present at 47°N during the *Cadoceras elatmae – Proplanulites koenigi* chrons of the early Callovian (Fig. 12), Metriorhynchidae indet. and Geosaurini indet. (morphotype 2) present at 50°N during the *Cadoceras elatmae* chron of the early Callovian (Fig. 12), and *Tyrannoneustes* sp. and cf. *Thalattosuchus* present at 46°N during the *Kosmoceras jason* chron of the middle Callovian (Fig. 12).

Together with stable isotope data, the presence of Metriorhynchidae indet., Geosaurini indet. (morphotypes 1 and 2), *Tyrannoneustes* sp. and cf. *Thalattosuchus* at 45–50°N during southernly range expansions of Boreal molluscs suggests that these species did indeed swim in moderate sea temperatures of approximately 13 °C, possibly as low as 5 °C. It is impossible to determine whether these metriorhynchids frequented northern latitudes during the warmer summer months or could survive throughout the year, although the rarity of metriorhynchid fossils from Russia possibly hints at the former hypothesis.

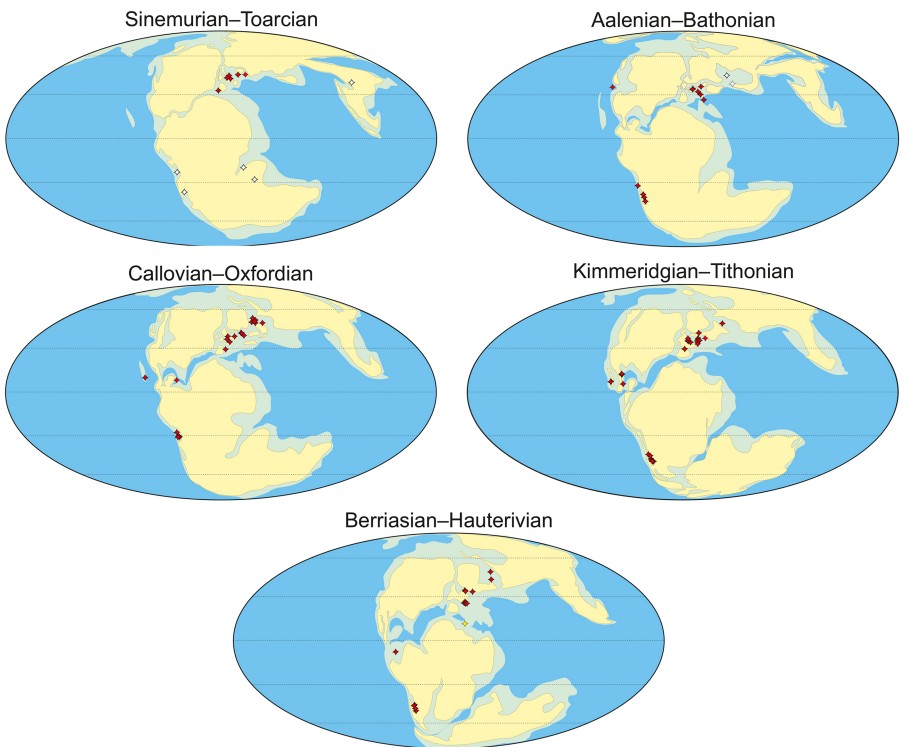

**Figure 12 Latitudinal distribution of metriorhynchoids during the Early Jurassic–Early Cretaceous.**
Red asterisks indicate metriorhynchoid localities, white asterisks indicate localities with indeterminate thalattosuchians and, for the Early Jurassic time interval; additionally teleosauroid localities from regions other than Europe; yellow asterisk at the Berriasian–Hauterivian paleogeographic map shows the youngest known metriorhynchid occurrence from the Barremian–Aptian (*Chiarenza et al., 2015*). For details on localities and palaeolatitude calculations see Table S1. Paleogeographic framework follows *Scotese (2004)* with modifications after *Lawver et al. (2020)*.

Amongst extant species, the genus *Alligator* is considered to be the most cold-tolerant (*Griggs & Kirshner, 2015*: 367). Alligators cope with prolonged cold periods primarily by taking refuge in dens or burrows (*Thorbjarnarson & Wang, 2010*; *Griggs & Kirshner, 2015*), with American alligators being able to recover from body temperatures of between 4–5 °C (*Brisbin Jr, Standora & Vargo, 1982*). One study of American alligators living in shallow ponds in South Carolina, USA, could survive living in water that ranged from 4.8–7.2 °C over winter (*Brisbin Jr, Standora & Vargo, 1982*). These animals made use of terrestrial basking behaviours and were consistently lethargic (*Brisbin Jr, Standora & Vargo, 1982*).

Given the metriorhynchid body plan (*Fraas, 1902*), building burrows is unlikely, and metriorhynchids could not engage in mouth-gape and osteoderm-mediated basking behaviours (*Young et al., 2010*). Moreover, it is unlikely that metriorhynchids could have survived being consistently lethargic in an open sea environment. Therefore, metriorhynchids being present in low-temperature environments is consistent with the hypothesis that they had an elevated metabolism (*Séon et al., 2020*).

Metriorhynchids are conspicuously absent from the high latitude part (= Arctic localities) of the Boreal realm. Other Mesozoic marine reptile groups, namely ichthyosaurs

and plesiosaurs, are known from high latitude deposits of the Boreal realm, from the Jurassic and the Early Cretaceous (*e.g.*, *Maxwell, 2010*; *Druckenmiller et al., 2012*; *Knutsen, Druckenmiller & Hurum, 2012a*; *Knutsen, Druckenmiller & Hurum, 2012b*; *Knutsen, Druckenmiller & Hurum, 2012c*; *Knutsen, Druckenmiller & Hurum, 2012d*; *Roberts et al., 2014*; *Roberts et al., 2020*; *Delsett et al., 2017*; *Zverkov et al., 2015*; *Zverkov, Grigoriev & Danilov, 2021*; *Rogov et al., 2019*; *Zverkov & Prilepskaya, 2019*; *Zverkov & Jacobs, 2021*). *Séon et al. (2020)* posited that metriorhynchids were poorly homeothermic endotherms, in contrast with the endo-homeothermy of ichthyosaurs and plesiosaurs (*Bernard et al., 2010*). The known fossil record of Metriorhynchidae is consistent with the contentions of *Séon et al. (2020)*, suggesting that metriorhynchids were unable to survive in the colder waters of the Boreal realm.

## CONCLUSIONS

The new specimens we describe herein allow us to expand the known geographical range of metriorhynchid taxa, with cf. *Thalattosuchus* present in the Middle Russian Sea during the early and middle Callovian, *Tyrannoneustes* present during the middle Callovian and *Torvoneustes* present during the middle Oxfordian. The enigmatic 'E'-clade metriorhynchids may also have been present in the Middle Russian seaway during the latest Oxfordian–early Kimmeridgian. We also referred two tooth crowns from the early Callovian to Geosaurini. These crowns are the oldest known specimens referred to Geosaurini; they exhibit unusual serrations morphologies and cannot be referred to any known clade solely on the basis of dental morphology. However, they show that: (1) contiguous well-defined microziphodonty had evolved in the Middle Jurassic, and (2) at least some metriorhynchid species had tooth crowns with both true microziphodonty and incipient microziphodonty.

Considering the findings described herein, the known latitudinal range of Metriorhynchidae is 50° North to 40–41° South. This range is based on specimens from the upper Bathonian to lower Callovian (Middle Jurassic) of Argentina and Russia. There is no evidence to suggest that metriorhynchids expanded their latitudinal range throughout their evolutionary history, as they are absent from Late Jurassic–Early Cretaceous horizons across the Boreal realm. Based on the specimens we describe herein, we know that metriorhynchids remained a component of the Middle Russian Sea fauna during the Late Jurassic and Early Cretaceous (up to 49° and 45° North, respectively).

Two of the tooth crowns described here were found in localities deposited during southernly range expansions of Boreal molluscs. Estimates suggest that sea surface temperatures were between 5 ° and 13 °C at that time. Therefore, these two specimens are the first evidence that metriorhynchids could frequent low-temperature environments. However, their rarity at latitudes above 45 degrees north and the absence of metriorhynchids in the Arctic and sub-Arctic areas, suggests that the specimens described here were at the northern-most potential range for this clade. This supports the contention (*Séon et al., 2020*) that metriorhynchids had an elevated metabolism but did not evolve an endo-homeothermic metabolism like ichthyosaurs and plesiosaurs.

**Institutional abbreviations**

| | |
|---|---|
| **GLAHM** | Hunterian Museum, Glasgow, UK |
| **KGM** | Kirov Geological Museum, Kirov, Russia |
| **MB** | Museum für Naturkunde, Berlin, Germany |
| **MRUM** | Mordovian Republican United Museum of local lore named after I. D. Voronin, Saransk, Russia |
| **NHMUK** | Natural History Museum, London, UK |
| **OUMNH** | Oxford University Museum of Natural History, Oxford, UK |
| **PETMG** | Peterborough Museum & Art Gallery, Peterborough, UK |
| **PIN** | Borissiak Paleontological Institute, Russian Academy of Sciences, Moscow, Russia |
| **SGM** | V.I. Vernadsky State Geological Museum of the Russian Academy of Sciences, Moscow, Russia |
| **SSU** | Saratov State University, Regional Museum of Earth Sciences, Saratov, Russia |
| **UPM** | Undory Paleontological museum, Undory, Ulyanovsk Region, Russia |
| **XKM** | Khvalynsk Museum of Local Lore, Khvalynsk, Saratov Region, Russia |

## ACKNOWLEDGEMENTS

We thank Andrey V. Stupachenko, who found PIN 5819/1, 5819/2, and SGM BX-12, Sergey I. Buganin (UPM, collected UPM 3031), Maxim S. Pichugin (UPM, collected UPM 3025), Arseny Grishin (collected UPM 3030) and Roman Rakitov (PIN) for his assistance with SEM photography. Dmitry V. Grigoriev and Alexander O. Averianov (Zoological Institute RAS) provided photographs of the vertebra UPM 1376. We also thank Oksana A. Kapitanova, chief curator of the Khvalynsk Museum of Local Lore, for providing information and photographs of the specimens under her care. We thank Alina Kanarkina for her help with the vector tracing of palaeogeographic maps. We also thank Mike Day (NHMUK) for collections access. Finally, we thank the three reviewers (Stéphane Hua, Michela Johnson, and Attila Ősi) for constructive input that helped improve the quality of this contribution.

### Funding

This work was supported by the Geological Institute of the Russian Academy of Sciences. The funders had no role in study design, data collection and analysis, decision to publish, or preparation of the manuscript.

### Grant Disclosures

The following grant information was disclosed by the authors:
Geological Institute of the Russian Academy of Sciences.

### Competing Interests

Mark T. Young is an Academic Editor for PeerJ.

## Author Contributions

- Mark T. Young conceived and designed the experiments, performed the experiments, analyzed the data, prepared figures and/or tables, authored or reviewed drafts of the article, and approved the final draft.
- Nikolay G. Zverkov conceived and designed the experiments, performed the experiments, analyzed the data, prepared figures and/or tables, authored or reviewed drafts of the article, and approved the final draft.
- Maxim S. Arkhangelsky conceived and designed the experiments, performed the experiments, analyzed the data, prepared figures and/or tables, authored or reviewed drafts of the article, specimen collection, and approved the final draft.
- Alexey P. Ippolitov conceived and designed the experiments, performed the experiments, analyzed the data, prepared figures and/or tables, authored or reviewed drafts of the article, specimen collection, and approved the final draft.
- Igor A. Meleshin analyzed the data, authored or reviewed drafts of the article, specimen collection, and approved the final draft.
- Georgy V. Mirantsev analyzed the data, authored or reviewed drafts of the article, specimen collection, and approved the final draft.
- Alexey S. Shmakov analyzed the data, authored or reviewed drafts of the article, specimen collection, and approved the final draft.
- Ilya M. Stenshin analyzed the data, authored or reviewed drafts of the article, specimen collection, and approved the final draft.

## Data Availability

The photographs of the fossil teeth and vertebrae are in the article.

## Supplemental Information

Supplemental information for this article can be found online at http://dx.doi.org/10.7717/peerj.15781#supplemental-information.

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
