# Peer review of "Thalattosuchian crocodylomorphs from European Russia, and new insights into metriorhynchid tooth serration evolution and their palaeolatitudinal distribution"

_PeerJ, doi:10.7717/peerj.15781_

## Round 0.1 · original submission · Minor Revisions

The study of thalattosuchians has resulted in a rich body of literature dating back to the mid-18th century. Please, try and avoid citing only the most recent papers.

I recommend you provide a table indicating the placement of the Russian stages Volgian and Ryazanian with respect to the ICS stages, the different formations your specimens come from, and the Tethyan and Boreal Ammonite zones. A map of European Russia indicating the different sites would also be useful.

As crocodylomorph teeth are notoriously difficult to identify, a table or a diagram showing the reasons for the identification of isolated teeth as Thalattosuchia indet., Metriorhynchidae indet., cf. Thalattosuchus, Tyrannoneustes sp., Geosaurini indet., and Torvoneustes sp. would be much appreciated.
In the reference list, I advise putting all the translated Russian titles in square brackets and adding “[in Russian]” after the page numbers.

Please, together with your unmarked revised manuscript, provide a marked-up copy as well as a document explaining how you have addressed every issue.

·

Basic reporting

34 : wrong, in North , South and central america, and already in russia….
46 this is not this has not been found that they are not existing.
48, please stop to be always focused only thalattosuchians!; Ellesmere (Norway), you have Eocene crocodiles and ectothermic
67 please in your bibliography, it seems nobody works until you, be modest…. Mateer, 1974, Buffetaut 1982, Hua, 1994, Buffetaut & hua 1997…..
71 take care with elevated metabolism, Romain and Séon have presented recently in a congress in Villers 2 years ago that in fact the “endothermy” could be localized and confirming Hua (1994) and histology….Their paper in on prep.
71 frankly, it seems like as usual that but that between 1902 and 2000 nothing happens… respect the other works
109 you missed buffetaut 1982 some remains in Dagestan
869 you haven’t taken in account Lemors et al. (2022) showing that it could be the same genus, this is not because this is not your result that you have not to take in account
882 no Buffetaut 1982, some people works before, take first hand paper…
883 please… same remarks stop to cite the last papers that has just cited previous works… respect the authors at least Buffetaut 1982, Fraas 1902, Arthaber , 1906; hua,, wenz, 1968, hua, 1994, Hua & buffetaut, 1997
920 check lemors (2022) all may be the same genus, this is not a wastebasket genus
938 you see lineage in tooth morphology without seeing variation intramandibular and ontogenic so your conclusions could be highly speculative especially with a phylogeny based only on your works
945 you cite lepage that is an amator, (and I hope not only because you cosigned this paper) and not E. Buffetaut that has worked nearly all his life on this same level in the French national CNRS scope…Please be more serious in your bibliography
970 please cite at least Muller Towe 2006 for german deposits that has made a great job and analysis and not one specimen = on new genus
995 Take care the apparent endemism is not a part because there are discoveries on particularly rich deposits studied from 2 centuries? look in Argentina, we discover a rich fauna. and it seems now that Tunisia and Marocco are going to deliver nice discoveries
1010 please cite Massare 1987 she was the first to speak about that!!!
999 very happy to see that nobody has worked before 2000, please this is not because you revised the taxinomy (and far to be recognised by everybody) that you have not to cite the others
1016 if you speak of range of living crocodylians why don’t you speak about Trans Oceanic Migration Hypothesis of the living genus Crocodylus (DUNSON, 1989) ? if you use actualism do it frankly
1031 surface current do not exist in the past?
1035 why don’t you compare to results in paleobiogeography of another marine mesosuchians, Dyrosaurids ? (Hua, 1996) this hypothesis is not so new…
1040 you mix species that are not in the same ammonite zone so separated from million years, so you assume that current and coast were unchanged during million years?
1074 this is isolated teeth : sure they were in place?.....
1088 no ! this is not because you have not found yet that they are rare or do not exist
1102 already said that take care with Seon because new results will confirm histology you are not using (Hua & De buffrenil, 1995) please take in account all data and confirming also Hua 1994
1112 please so explain Eocene crocodile from Ellesmere, please look the other families in your conclusions
1119 frankly speaking if you are unable to assign correctly in taxonomy, this is not conform to zoological code so invalid
1127 geographic morph no?
1130 again this is not because you haven’t found that it not exist
1140, please this is not the first… sorry first hua et al. 1998 in your bibliography!
1143 please integrate concrete data (hua & de buffrenil, 1995) to have an objective conclusion.
1143 so on isolated teeth that could be removed in sediments, not in place, and you are unable to determine, and assuming a constant currentology and Temperature across the column of water you extrapolate on partial bibliography they are rare and not fully endothermic : your conclusions are too fragile.
Again check Ellesmere!
from the same area (you seems to forget) Hua et al. 1998 described more complete specimens that have best chances to be in place, so where it could be possible to conclude and to be careful in conclusions, you are going to far in your conclusions

In your Table : not ammonite zone? , this is dangerous to compare asynchronous specimens

Experimental design

identified gaps in bibliography that could altered strongly conclusiond

Validity of the findings

At first, this is not the first discovery in this area (Hua, et al. 1998) and more over on more complete remains : here this is just isolated tooth, that could be not in place, with an uncertain determination.
From there the authors push very far (too far?) their conclusions despite some weakness in bibliography (details below)

From paleontological point of view :
*it lacks some essential previous works to privelege only recents works (and even sometimes new papers), please respect the previous work to avoid autocitation
* to have a better vision of paleolatitude in fossil crocodylians, the authors have to check the others families with Ellemere deposits or even paleogeography of Dyrosaurids (like Hua, 1996)
* the taxonomy used being foggy and not even recently recognized
* for thermical paleobiology only isotopic data (cosigne by the author) have been used and not histological data that are not going on the way detailed by authors (Hua, de Buffrenil, 2010).

from actualism point of view:
speaking of repartition of living crocodylians without speaking and comparing to TOMH hypothesis developped since 1989 fon interpreting the repartion of Crocodylus is a gap for me.

from geological view : comparing fossils from various ammonite zone, assuming a kind of fixism in paleogeography, is a bit dangerous

Additional comments

Enhance the bibliography with not only recent papers and not only from thalattosuchians.
Take care of your conclusion this is not the first finding in this area and do not push too far your conclusions, especially with poor remains, without integrating the maximum of various data and not only on thalattosuchians

·

Basic reporting

An overall clear objective; professional English throughout; good literature review and references.

One suggestion: an additional figure highlighting where the localities/quarries are, with an accompaning stratigraphic column, would be beneficial for the geological settings section.

Experimental design

Experimental design is valid and offers information on a knowledge gap within metriorhynchids.

- Mention the use of SEM in the methods section.

Validity of the findings

Interesting findings, good anatomical descriptions. Conclusions are well stated.

- For specimens PIN 5819/1/PIN 5819/2/PIN 5819/6, PIN 5819/7, it might be good to have a line or two for those unfamiliar with thalattosuchian dental morphology explaining why these are assigned to Metriorhynchoidea (= Metriorhynchidae indet.) and not Teleosauroidea (mainly due to their indeterminate status).

Additional comments

- Are the palaeolatitude calculations present in Table 1? I see the locality but not the coordinates (unless that is what the authors mean).

- Line 231: "break" instead of "broke".

- Lines 460-464 and lines 505-509 are copied word-for-word with one another. Maybe use a couple of different synonyms to change them a bit, it was a bit jarring for me to read.

·

Basic reporting

The manuscript is professionally written in English. The authors are absolutely aware of the literature on thalattosuchian crocodyliforms, the fossil material and any unresolved questions and uncertainties associated with the group. This is supported by the relevant literature cited in the appropriate places.
The illustrations are clear and attractive, and the relevant morphological characters used in the description are clearly visible.
The main chapters of the manuscript are basically fine, but the structure of the "Systematic Palaeontology" chapter is a bit hard for me to follow.
E.g. from Metriorhynchidae indet. there is:
Metriorhynchidae indet. Morphotype 1 (l. 271)
Metriorhynchidae indet. Morphotype 2. (l. 316)
Metriorhynchidae indet. (l. 805)
Metriorhynchidae indet. (l. 858)
The last two taxa are based on vertebrae, so it is not possible to compare them with Morphotype 1 and 2. but perhaps these vertebrae could be discussed in just one Metriorhynchidae indet. block.

I would expect a 'systematic' discussion of each fossil, but here it seems to be in chronological order. Perhaps the genus level specimens should be described first and move towards increasingly uncertain classification.

Otherwise, the manuscript contains very valuable information on fossil crocodliforms and, after some revisions, I would strongly support its publication.

Experimental design

This manuscript documents a few isolated crocodilian teeth and vertebrae from the European part of Russia, from the Middle Jurassic to Lower Cretaceous rocks. The finds are thought by the authors to belong to the thalattosuchian group. Both the morphological features of the teeth and the vertebrae are carefully circumscribed and compared with other finds.
The authors are clear about the significance of the finds: these specimens come from an area where this group has been almost entirely undocumented, a paleo-area that in the Jurassic and Cretaceous periods hosted the Middle Russian Sea, which connected the northern boreal waters with the southern Tethys Ocean. This also raises paleoecological questions about the paleolatitude distribution of the Thalattosuchia group, which has been scarcely studied. However, this new material adds much new information to the question.

Validity of the findings

The conclusions that can be drawn from the fossil record described here are moderate and well-founded.

Additional comments

Small comments:

- "Geological settings" contains many information on the paleogeographical background of the area. This is very useful as it gives a temporal, spatial and oceanic framework for the fossils described herein. So I suggest that the heading of this chapter might be better somehow as "Geological settings and paleogeography".

- In addition a geographic map from the area with the localities would be very useful to see not only the temporal but recent spatial distances between the localities.

- line 290: the authors mention taphonomic wear for the teeth figured on Fig 2. However, on Fig 2N-R the apex of that tooth is slightly concave on the dentine which is most probably the result of dental wear (i.e. wear due to feeding). (Abrasion due to transportation or other taphonomical processes do not make similar wear without eroding the carinae or other part s of the crown).

- The authors write in line 633: "Finally, we cannot refer it to Dakosaurus as the crown lacks the characteristic ‘robust’ Dakosaurus morphology, ‘carinal flanges’, carinal macrowear facets..."My opinion is that macrowear facets is not a stable taxonomical charater since in replacement or newly erupted teeth which were not in use and still unworn we won't see any kind of wear facet.

---

## Round 0.2 · accepted · Accept

I confirm that your manuscript has been accepted for publication.